# Nlrc3 signaling is indispensable for hematopoietic stem cell emergence via Notch signaling in vertebrates

Shuyang Cai[1,2,3,4,5,12], Honghu Li[1,2,3,4,12], Ruxiu Tie[1,2,3,4,6,7,12], Wei Shan[1,2,3,4,12], Qian Luo[1,2,3,4,12], Shufen Wang[8,12], Cong Feng[9,10], Huiqiao Chen[8], Meng Zhang[1,2,3,4], Yulin Xu[1,2,3,4], Xia Li [1,2,3,4], Ming Chen [9,10], Jiahui Lu[5] ✉, Pengxu Qian [2,3,4,11] ✉ & He Huang [1,2,3,4] ✉

Hematopoietic stem and progenitor cells generate all the lineages of blood cells throughout the lifespan of vertebrates. The emergence of hematopoietic stem and progenitor cells is finely tuned by a variety of signaling pathways. Previous studies have revealed the roles of pattern-recognition receptors such as Toll-like receptors and RIG-I-like receptors in hematopoiesis. In this study, we find that *Nlrc3*, a nucleotide-binding domain leucine-rich repeat containing family gene, is highly expressed in hematopoietic differentiation stages in vivo and vitro and is required in hematopoiesis in zebrafish. Mechanistically, *nlrc3* activates the Notch pathway and the downstream gene of Notch *hey1*. Furthermore, NF-kB signaling acts upstream of *nlrc3* to enhance its transcriptional activity. Finally, we find that *Nlrc3* signaling is conserved in the regulation of murine embryonic hematopoiesis. Taken together, our findings uncover an indispensable role of *Nlrc3* signaling in hematopoietic stem and progenitor cell emergence and provide insights into inflammation-related hematopoietic ontogeny and the in vitro expansion of hematopoietic stem and progenitor cells.

Hematopoietic stem and progenitor cells (HSPCs) have the extraordinary ability to both self-renew and generate all the known blood cell lineages. In all vertebrate embryos, HSPCs arise from the hemogenic endothelium (HE) of the dorsal aorta (DA) during a transitory developmental process as the hemogenic endothelial to hematopoietic transition (EHT). Previous studies have provided a more comprehensive understanding of the signaling pathways that orchestrate HSPCs emergence. Pattern-recognition receptors (PRRs), including Toll-like receptor 4 (*TLR4*) and RIG-I-like receptors (RLR)[1,2], play essential roles in embryonic HSPC production. However, it remains largely unknown what the roles and mechanisms of the nucleotide-binding oligomerization domain (NOD)-like receptors (NLRs) are in this process. It is known, however, that NLRs function as sensors of the innate immune system and regulators of

[1]Bone Marrow Transplantation Center, the First Affiliated Hospital, Zhejiang University School of Medicine, Hangzhou, China. [2]Liangzhu Laboratory, Zhejiang University Medical Center, Hangzhou, China. [3]Institute of Hematology, Zhejiang University, Hangzhou, China. [4]Zhejiang Engineering Laboratory for Stem Cell and Immunotherapy, Hangzhou, China. [5]Shanghai Municipal Hospital of Traditional Chinese Medicine, Shanghai University of Traditional Chinese Medicine, Shanghai, China. [6]Department of Hematology, the Second Clinical Medical College, Shanxi Medical University, Taiyuan, China. [7]Department of Hematology, Taizhou Hospital of Zhejiang Province Affiliated to Wenzhou Medical University, Linhai, China. [8]Sir Run Run Shaw Hospital, Zhejiang University School of Medicine, Hangzhou, China. [9]Department of Bioinformatics, College of Life Sciences, Zhejiang University, Hangzhou, China. [10]Bioinformatics Center, The First Affiliated Hospital, School of Medicine, Zhejiang University, Hangzhou, China. [11]Center for Stem Cell and Regenerative Medicine and Bone Marrow Transplantation Center of the First Affiliated Hospital, Zhejiang University School of Medicine, Hangzhou, China. [12]These authors contributed equally: Shuyang Cai, Honghu Li, Ruxiu Tie, Wei Shan, Qian Luo, Shufen Wang. ✉e-mail: lujiahui_sci@126.com; axu@zju.edu.cn; huanghe@zju.edu.cn

cytokine secretion, which are important for vertebrate hematopoiesis.

*Nlrc3*, an intracellular member of the NLR family, has been reported to be highly expressed in various immune cells[3,4]. In mammals, *Nlrc3* has been shown to be a negative regulator of numerous cellular populations, such as macrophages, dendritic cells (DCs), and T lymphocytes, and has also been shown to be involved in the host innate immune response to intracellular DNA and DNA viruses through NF-kB signaling[5–9]. Furthermore, *Nlrc3* deficiency has been shown to increase the expression of proinflammatory cytokines, including interleukin-1β (*il1b*), *il6*, *il12* and tumor-necrosis factor a (*TNF-α*) in lipopolysaccharide-induced macrophages[7]. In teleost, emerging evidence has suggested the regulatory roles of piscine *NLRC3*-like genes in pathogen infections and the production of proinflammatory cytokines[10,11]. Among these diverse biological functions that may affect homeostasis or disease pathogenesis, the explicit role of *Nlrc3* in HSPCs, particularly in embryonic stages, remains poorly understood.

Canonical Notch signaling is essential in the regulation of definitive HSPCs[12]. The emergence of the hemogenic endothelium (HE), from which definitive HSPCs are derived, is concomitant with arterial differentiation. Notch has been shown to be responsible for the loss of arterial identity[13,14]. In particular, proinflammatory factors such as *tnfa*[15], interferon-gamma (*IFN-γ*)[16], and *il6*[17] have been reported to be associated with Notch in regulating HSPCs. However, the regulatory relationship between *Nlrc3*, one of the PRRs involved in regulating these factors, and Notch remains undocumented. Moreover, among the ligands and receptors of Notch signaling, the Hey genes, which are helix-loop-helix transcription factors, such as *Hey2*, act synergistically during all stages of arterial establishment[18]. Although it has been reported that *Hey1* is involved in vascular development, little is known about its role in HSPC emergence[19,20].

The transcription factors of the nuclear factor k-light-chain enhancer of the activated B core (NF-kB) family serve as the master controllers of the inflammatory response, and the conserved NF-kB–Notch pathway is essential in HSPC emergence[1,21]. NLR proteins such as *NOD1*, *NOD2*, and *NLRP3* can positively alter the activation of NF-kB in response to peptidoglycan stimulation[22–25], but *Nlrc3* has been reported to be an inhibitor of NF-kB through modulation of the ubiquitination of TRAF6[7], which has been shown to suppress the expansion of HSPCs in the fetal liver[26]. These results prompted us to hypothesize that *Nlrc3* might play critical roles in vertebrate embryonic hematopoiesis through interacting with NF-kB.

In this study, we delineated the functional roles of *Nlrc3* signaling in the HSPC emergence in various systems, including published data of single-cell RNA-seq (scRNA-seq) and the embryonic stem cell (ESC) differentiation system in vitro[27–30], as well as in zebrafish and mouse models in vivo. Mechanistically, we found that during the embryonic stage, NF-kB regulates *Nlrc3* through transcriptional activity, and *Nlrc3* acts upstream of Notch and *Hey1*, these factors forming an essential axis of NF-kB-*Nlrc3*-Notch-*Hey1* is indispensable for HSPCs emergence. Overall, our study revealed an undiscovered role of *Nlrc3* signaling in the generation of embryonic HSPCs under nonpathogenic conditions and emphasizes the vital role of pattern recognition receptor signals in the formation of HSPCs during development.

## Results

### *NLRC3* is highly expressed during vertebrate HSPC development in vivo and in vitro

Previous studies in vertebrate embryos uncovered the role of inflammatory signaling in HSPC emergence. To explore the potential pathways involved in hematopoietic ontogeny and differentiation in this stage, we screened the dynamic expression of NLR family genes and other genes related to inflammation in the published scRNA-seq data in vertebrates' embryos during HSPC development. In a data profiling caudal hematopoietic tissue (CHT) of zebrafish at 3.5 and 4.5 days

postfertilization (dpf)[27], Compared to many other NLR family paralogs, including *nlrp1*, *nlrp3*, *nlrc3l*, *nlrc5*, *nlrc6*, *nlrx1* and *nlrp16*, the expression of *nlrc3* is relatively higher in the ECs and HSPCs subpopulations (Fig. 1a). To further investigate the expression of *nlrc3* in mammalian models, we analyzed published single-cell transcriptomic data profiling mouse embryo from 9.5 days post coitus (dpc) to 11.5 dpc[28]. Comparison of Aorta Gonad Mesonephros (AGM)−derived HSPCs and fetal liver (FL)-derived HSPCs with venous endothelial cells (VECs), arterial endothelial cells (AECs), pre-hemogenic endothelial cells (pre-HECs) and hemogenic endothelial cells (HECs), the expression of *Nlrc3* increases with hematopoietic maturation, especially when comparing HECs with HSPCs, which is the stage that hemogenic endothelial to hematopoietic transition (EHT) occurs (Fig. 1b). And the expression of *Nlrc3* is not only highly expressed among the NLR family genes but also like that of *Tlr4*, which has been reported to be involved in the regulation of HSPC development[1]. Furthermore, in the latest single-cell transcriptome map of human hematopoietic tissues from the first trimester[29], the expression trend of *NLRC3* was also validated to be enriched in venous endothelium (VE), arterial endothelium (AE), hemogenic endothelium (HE), HSCs and HPCs which are defined as Non-HSCs (Fig. 1c), especially the differentiation stage from HE to HSCs.

Although rather in the primitive stage, the hematopoietic differentiation system using embryonic stem cells (ESCs) is a convenient research model that simulates and observes the in vivo embryonic hematopoiesis, in which cells gradually differentiate into the mesoderm, HE, and HSPCs stages, and are accompanied by specific different molecular markers at different stages[31,32]. To further verify the expression of *Nlrc3* in vitro, we compared FL- and bone marrow (BM)-derived HSPCs with HSC-like cells derived from mouse ESCs (Supplementary Fig. 1a)[30]. Differential gene expression analysis using RNA-seq revealed that among the NLR family genes, *Nlrc3* expression was highly enriched with the expression of genes that have been reported to regulate the emergence of HSCs, including *Tlr4*[1] and *IFN-γ*[16] (Supplementary Fig. 1b).

We further validated our results by employing the in vitro HSC-like cell differentiation system using human embryonic stem cells (hESCs) (Supplementary Fig. 1c). We examined the dynamic change of *NLRC3* during the 12-day hematopoietic differentiation process by qPCR and found that compared with day 0, the expression of *NLRC3* gradually increased along with the hematopoietic differentiation from hESCs (Supplementary Fig. 1d). Moreover, we isolated cells from different stages (hESCs at Day 0, CD309+ mesoderm cells at Day 3, CD31+CD34+ HE cells at Day 6, CD43+ HSPCs cells at Day 9, and CD45+ HSPCs at Day 12). Our qPCR results showed that, compared with undifferentiated hESCs, the expression of *NLRC3* in HE was significantly increased by approximately 300-fold and was even higher in CD43+ and CD45+ HSPCs (Supplementary Fig. 1e). *TLR4* and *NLPR3*, which have been reported to be regulators of HSPCs emergence and differentiation, showed a similar trend to *NLRC3* (Supplementary Fig. 1f, g). Overall, these results suggest that *NLRC3* was highly expressed during embryonic HSPC development in vivo and vitro and might play an important role in HSPC emergence in vertebrates.

### *Nlrc3* signaling is indispensable for HSPC production in zebrafish

To investigate whether *nlrc3* signaling is required for in vivo HSPC emergence, we synthesized probes from the full-length mRNA of *nlrc3* and utilized them to observe the in-situ expression of *nlrc3* in zebrafish at different developmental stages. The WISH experiment demonstrated that *nlrc3* was expressed from the 1-cell stage and, importantly, showed specific expression in the AGM region of zebrafish at 24–28hpf, which coincides with the onset of HSPC generation. There was also a high expression of *nlrc3* at 72hpf in the CHT region, which follows a similar spatiotemporal expression pattern as HSPC generation, indicating that *nlrc3* may be involved in the development of

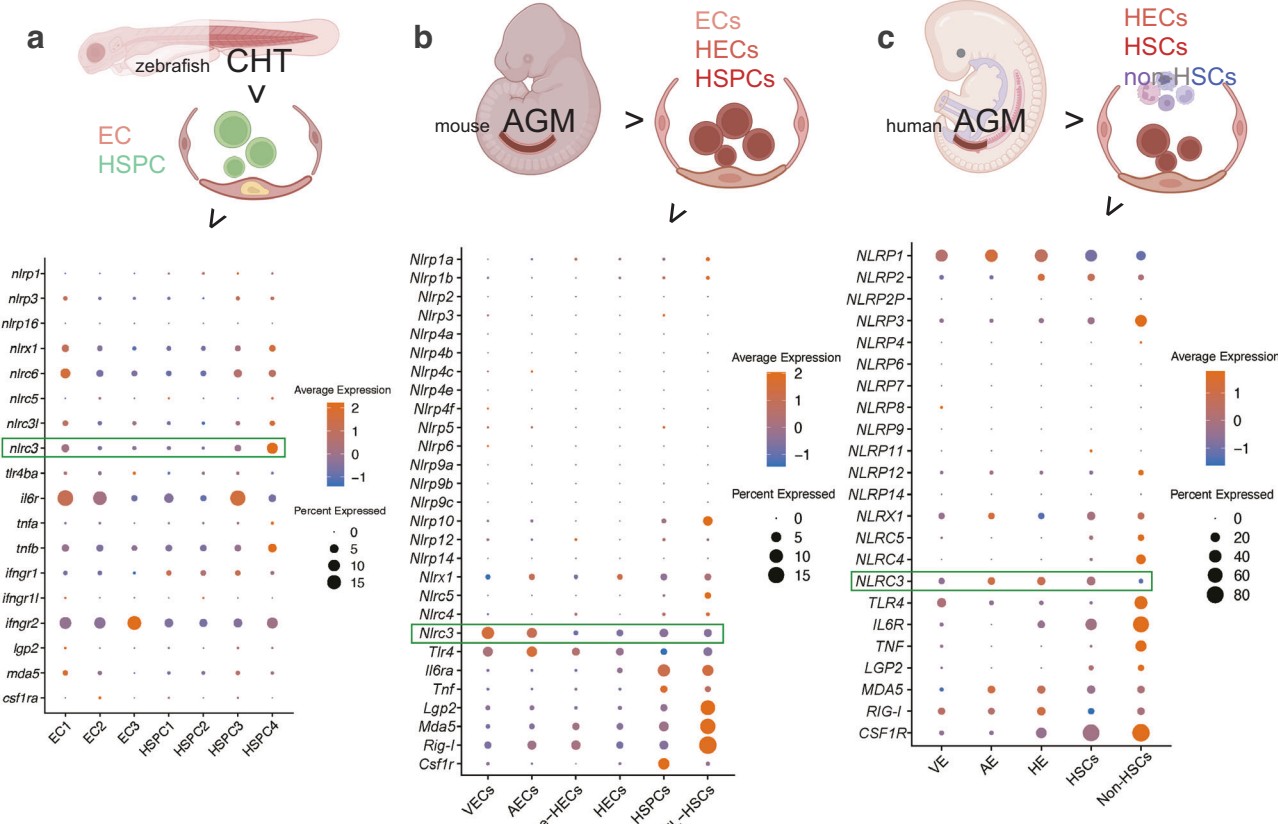

**Fig. 1 | *NLRC3* is highly expressed during vertebrates HSPC development.**
**a** Schematic paradigm of zebrafish caudal hematopoietic tissue (CHT) tissue for single-cell RNA-seq (scRNA-seq) profiling. Bubble plot of scRNA-Seq data demonstrating the expression of nucleotide-binding domain leucine-rich repeat (NLR) family members, including *Nlrc3* and related inflammatory genes in the clusters of endothelial cells (ECs) and Hematopoietic stem and progenitor cells (HSPCs).
**b** Schematic paradigm of mouse embryonic Aorta Gonad Mesonephros (AGM) for scRNA-seq profiling. Bubble plot of scRNA-Seq data demonstrating the expression of NLR family members, including *Nlrc3* and related inflammatory genes in the

clusters of venous endothelial cells (VECs), arterial endothelial cells (AECs), pre-hemogenic endothelial cells (pre-HECs), hemogenic endothelial cells (HECs), HSPCs, fetal liver (FL)-derived HSPCs. **c** Schematic paradigm of human embryonic AGM for scRNA-seq profiling. Bubble plot of scRNA-Seq data demonstrating the expression of NLR family members, including *NLRC3* and inflammatory genes in the clusters of venous endothelium (VE), arterial endothelium (AE), hemogenic endothelium (HE), HSPCs and Non-HSCs. Illustrations created with BioRender.com. Source data are provided as a Source Data file.

HSPCs (Supplementary Fig. 2a). By co-staining with arterial endothelial-specific markers *dlc* and *efnb2* (Supplementary Fig. 2b, c), hematopoietic stem/progenitor cell-specific markers *runx1* and *cmyb* (Supplementary Fig. 2d, e), we found that co-staining *nlrc3* with these probes can effectively enhance the expression of HE and HSPCs. These results suggest that *nlrc3* may be involved in the regulation of this process in zebrafish.

In order to validate this hypothesis, loss-of-function experiments were performed using zebrafish with a targeted Morpholino (MO) (Supplementary Fig. 3a). We observed the floor of the DA region in Tg (*runx1*:EGFP/ *kdrl*:mCherry) double-transgenic embryos at 28 hpf by confocal microscopy, *runx1* is a conserved HSPC marker and *kdrl* is a vascular endothelial-specific marker, the *runx1⁺kdrl⁺* cells signifies HSPCs occurs through the hemogenic endothelial (HE) to hematopoietic transition (EHT) process is this time point, and the number of *runx1⁺kdrl⁺* HSPCs of morphants was significantly lower than that in control embryos (Fig. 2a, b). By using the Tg(*cmyb*:EGFP/ *kdrl*:mCherry) double-transgenic line at 48 hpf, *cmyb* is another conserved HSPC marker, we found that the number of *cmyb⁺kdrl⁺* HSPCs of morphants within the ventral wall of the dorsal aorta (VDA) markedly decreased in *nlrc3* morphants (Fig. 2c, d). Tg (*CD41*: GFP), which is a well-established transgenic line with HSPC expansion in the CHT, also had reduced expression in morphants at 72 hpf (Fig. 2e, f). Furthermore, we performed whole-mount in situ hybridization

(WISH) to measure the expression of *runx1* and *cmyb*, which are nascent HSPC markers at early developmental stages, and found that the expression of *runx1* and *cmyb* was significantly reduced at 28 hpf and at 36 hpf, respectively, in the aortic floor of *nlrc3* morphants compared with their wild-type siblings (Fig. 2g, j). To further demonstrate the role of *nlrc3* in HSPC emergence, we generated *nlrc3* mutant by the CRISPR/Cas9, 29 bp base is knocked out in homozygotes (Supplementary Fig. 2b, c). The expression of *nlrc3* was abrogated in mutants at the site of HSPC occurrence at 24–36 hpf (Supplementary Fig. 3d, e). Consistent with previous results, the expression of *runx1* and *cmyb* was decreased in mutants by WISH at 28 hpf and 36 hpf, respectively (Fig. 2g–j). With this mutant, we also validated the downregulation of hematopoietic markers and *nlrc3* at 28 hpf by qPCR (Fig. 2k). The similar phenotype between morphants and mutants allows for the experimental design to be tailored based on specific needs. To obtain more comprehensive evidence, we examined *gata2b*, a critical early hematopoietic marker[12]. Through qPCR, we found a downregulation of *gata2b* in the mutant at 28 hpf. Therefore, we synthesized the full-length mRNA of *gata2b* and overexpressed it. The reduction in *runx1* at 28hpf and *cmyb* at 36 hpf due to *nlrc3* deficiency was partially restored by the rescue of *gata2b* (Supplementary Fig. 3f, g). This result was also confirmed by qPCR analysis (Supplementary Fig. 3h). These results indicated that *nlrc3* was required for HSPC emergence and expansion.

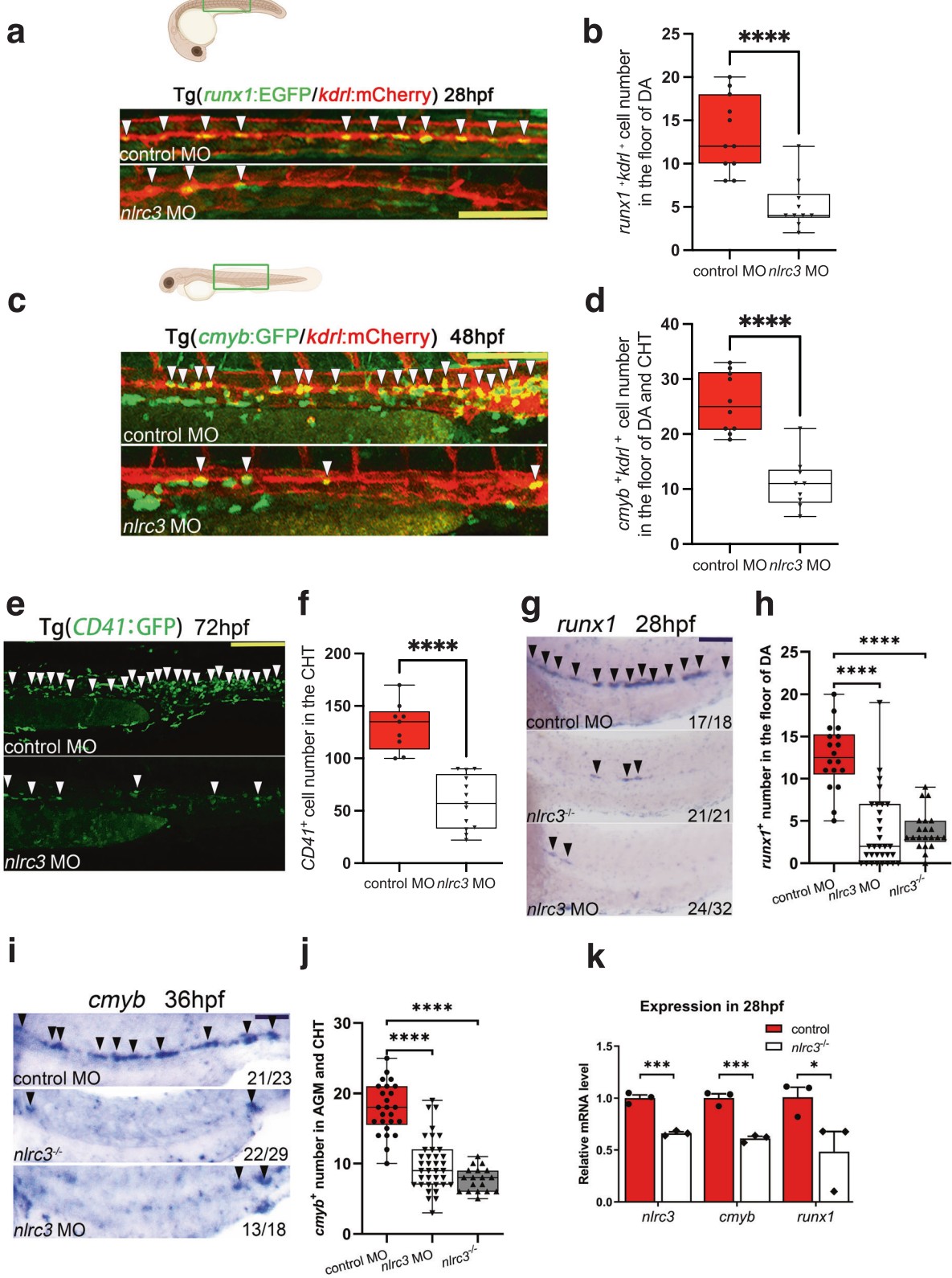

To further evaluate the effect of *nlrc3* on hematopoietic differentiation, we examined the expression of *rag1*, which specifically indicates the differentiation capability of HSPCs for T cell lineages. WISH results showed that *rag1* expression was markedly reduced in both morphants and mutants (Fig. 3a). A similar phenotype was observed in Tg (*lck*:EGFP) transgenic morphants at 120 hpf (Fig. 3b), while lymphoid genes, including *ikaros*, *rag1*, *lck*, and *il7r*, were

downregulated in mutants, as assessed by qPCR (Fig. 3c). However, the expression of the thymic epithelial cell marker *foxn1* was normal (Fig. 3a), suggesting that T-cell defects in these deficient embryos resulted from early HSPC defects. The expression of *gata1a*, an erythrocyte-specific marker, was decreased in the posterior blood island (PBI) and trunk regions of morphants at 36 hpf and 96 hpf, respectively (Supplementary Fig. 4a, Fig. 3d), which was further

**Fig. 2 | *Nlrc3* signaling is indispensable for HSPC production in zebrafish.**
**a, b** Confocal imaging showing the number of hemogenic endothelium and
emerging HSPCs in *runx1*⁺*kdrl*⁺ cells from Tg (*runx1*:EGFP/*kdrl*:mCherry) embryos at
28 hpf in the AGM (white arrowheads) in control embryos and *nlrc3* morphants with
quantification (**b**). ****P < 0.0001, *n* = 11, 10 embryos. **c, d** Confocal imaging showing
the number of *cmyb*⁺*kdrl*⁺ cells in Tg (*cmyb*:EGFP/*kdrl*:mCherry) embryos at 48 hpf
in the AGM (white arrowheads) in control embryos and *nlrc3* morphants with
quantification (**d**). ****P < 0.0001, *n* = 10, 9 embryos. **e, f** Confocal imaging showing
the number of HSPCs in Tg (*CD41*:GFP) embryos at 72 hpf in the CHT (white
arrowheads) in control embryos and *nlrc3* morphants with quantification (**f**).
****P < 0.0001, *n* = 9, 13 embryos. **g, h** Expression of the HSPC marker *runx1* in *nlrc3*
morphants and mutants in the AGM region at 28 hpf by whole mount in situ
hybridization (WISH) (black arrowheads) with quantification (**h**) ****P < 0.0001,

*n* = 18, 32, 21 embryos. **i, j** Expression of the HSPC marker *cmyb* in *nlrc3* morphants
and mutants in the region of AGM and CHT at 36 hpf by WISH (black arrowheads)
with quantification (**j**) ****P < 0.0001, *n* = 25, 37, 18 embryos. **k** Expression of *nlrc3*
and the HSPC genes *runx1*, *cmyb* in control embryos and *nlrc3* morphants at 28 hpf
by qPCR. *P = 0.0462, ***P = 0.0006, 0.0010, *n* = 3 biological replicates. Error bars,
mean ± s.d., ****P < 0.0001, by using two-tailed, unpaired Student's *t*-test in
(**b, d, f, k**), one-way ANOVA – Sidak test in (**h, j**). For the box plots in (**b, d, f, h, j**), box
limits extend from the 25th to 75th percentile, while the middle line represents the
median. Whiskers extend to the largest value no further than 1.5 times the inter-
quartile range (IQR) from each box hinge. Scale bars, 100 μm in (**a, c, e, g, i**).
Illustrations created with BioRender.com. Source data are provided as a Source
Data file.

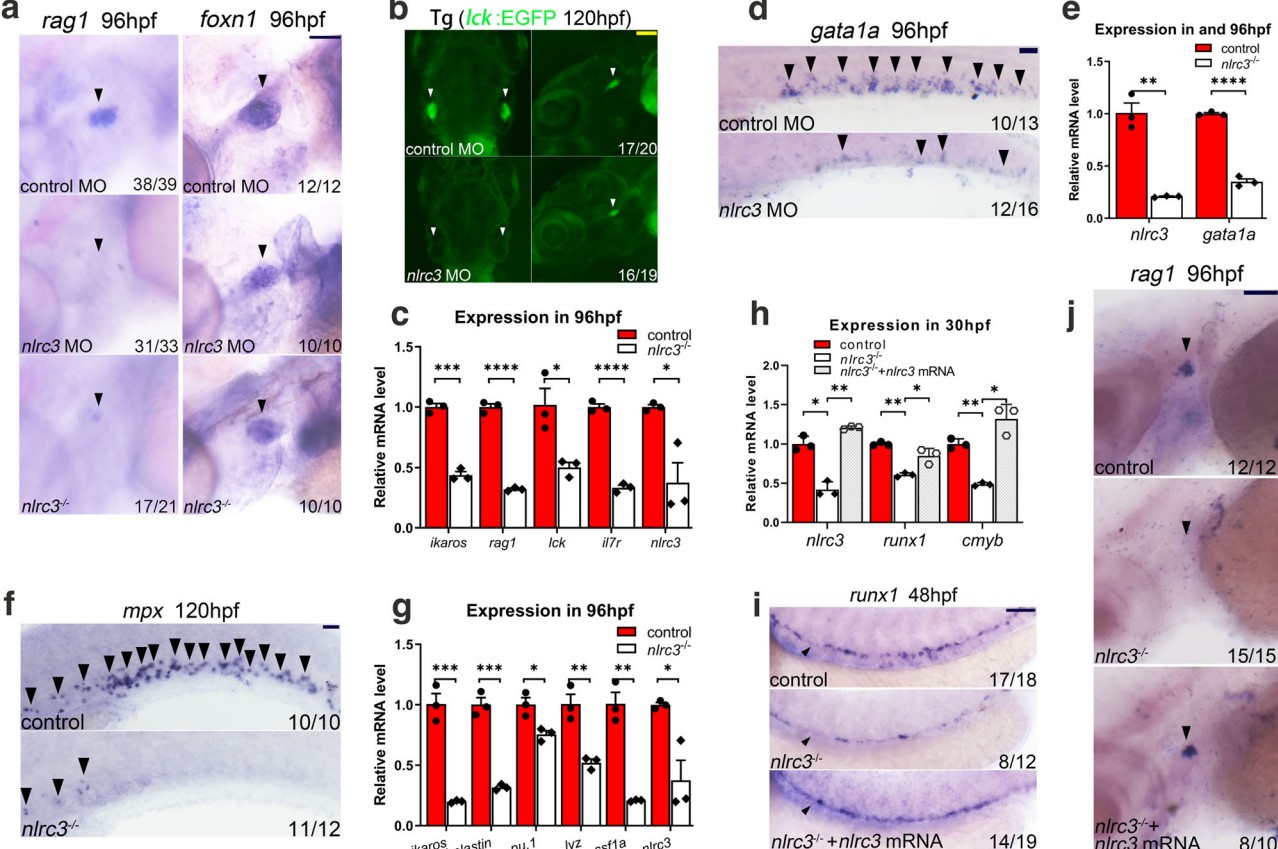

**Fig. 3 | *Nlrc3* signaling is imperative for HSPC differentiation in zebrafish.**
**a** Expression of the lymphoid marker *rag1* in *nlrc3* morphants and mutants and the
thymic epithelial cell marker *foxn1* in the thymus of control embryos, *nlrc3* mor-
phants, and mutants at 96 hpf (black arrowheads). **b** Snapshot of *lck*⁺ cells in Tg
(*lck*:EGFP) embryos in the thymus of control embryos and *nlrc3* morphants at 5 dpf
(white arrowheads). **c** Expression of *nlrc3* and the lymphoid genes *ikaros*, *rag1*, *lck*,
and *il7r* in control embryos and *nlrc3* morphants at 96 hpf by qPCR. *P = 0.0229,
0.0197, ***P = 0.0009. **d** Expression of the erythroid marker *gata1a* in control
embryos and *nlrc3* morphants by WISH at 96 hpf (black arrowheads). **e** Expression
of *nlrc3* and the erythroid gene *gata1a* in control embryos and *nlrc3* morphants at
96 hpf by qPCR. **P = 0.0011. **f** Expression of the myeloid marker *mpx* in control and
*nlrc3* morphants by WISH at 120 hpf (black arrowheads). **g** Expression of *nlrc3* and
the myeloid gene *ikaros, l-plastin, pu.1, lyz,* and *csf1a* in control embryos and *nlrc3*
mutants at 96 hpf by qPCR. *P = 0.0178, 0.0197, **P = 0.0049, 0.0011, ***P = 0.0007,

0.0003. **h** Expression of *nlrc3* and the HSPC genes *runx1* and *cmyb* in control, *nlrc3*
morphants, and *nlrc3* morphants with overexpression of *nlrc3* mRNA at 30 hpf by
qPCR. *P = 0.0286, 0.0272, 0.0158, **P = 0.0040, 0.0047, 0.0068. **i** Expression of
the HSPC marker *runx1* in control embryos, *nlrc3* morphants, and *nlrc3* morphants
with overexpression of *nlrc3* mRNA at 48 hpf by WISH (black arrowheads).
**j** Expression of the lymphoid marker *rag1* in control embryos, *nlrc3* morphants, and
*nlrc3* morphants with overexpression of *nlrc3* mRNA at 96 hpf by WISH (black
arrowheads). Error bars, mean ± s.d., *P < 0.05, **P < 0.01, ***P < 0.001,
****P < 0.0001, the exact *p*-values mentioned above are listed from left to right, by
using two-tailed, unpaired Student's *t*-test in (**c, e, g**). Two-way ANOVA with Tukey's
post hoc test in (**h**). *n* = 3 biological replicates. Scale bars, 100 μm in (**a, b, d, f, i, j**).
Illustrations created with BioRender.com. Source data are provided as a Source
Data file.

verified by qPCR analysis in mutants (Fig. 3e). We also examined the
phenotypes of Tg (*lcr*:EGFP) and Tg (*gata1a*:DsRed) transgenic mor-
phants at 60 hpf (Supplementary Fig. 4b, c). These two transgenic lines
are capable of tracking erythrocytes in blood vessels, and the results
strongly suggest that *nlrc3* signaling is essential for definitive

erythropoiesis. Myeloid lineage differentiation was also impaired in
mutants, as assayed by *mpx* at 96 hpf and *l-plastin* at 120 hpf by WISH
(Fig. 3f, Supplementary Fig. 4f). Similar phenotypes were assessed in
Tg (*mpx*:EGFP) and Tg (*lyz*:DsRed) transgenic morphants as the neu-
trophil marker at 60 hpf and 72 hpf, respectively, in the CHT region

(Supplementary Fig. 4d, e), and the expression of myeloid-lineage genes in mutants at 96 hpf was found to be decreased by qPCR (Fig. 3g). These results indicated that HSPC differentiation was impaired when *nlrc3* signaling was disturbed.

To determine if *nlrc3* was sufficient for embryonic hematopoiesis, we synthesized *nlrc3* mRNA for rescue experiments. The decreased expression of *runx1* at 48 hpf and *rag1* at 96 hpf could be rescued by overexpression of *nlrc3* mRNA, as assessed by WISH, and qPCR also validated this trend at 30 hpf (Fig. 3h–j). These results demonstrated that HSPC defects in *nlrc3*-deficient embryos were specific.

To further examine the role of *nlrc3* in the first wave of hematopoiesis, which is generally called the "primitive" wave, we performed WISH to measure the expression of *gata1a* at 26 hpf (Supplementary Fig. 5a). qPCR of *gata1a* at 30 hpf verified this phenotype, confirming that *nlrc3* signaling may not impact primitive erythropoiesis (Supplementary Fig. 5b). However, the expression of *mpx* was decreased at 24 hpf, as measured using WISH (Supplementary Fig. 5c), indicating that the development of primitive neutrophils depended on *nlrc3* signaling. Additionally, the expression of *mfap4*, a specific marker of macrophages, was reduced in *nlrc3*-deficient embryos at 25 hpf (Supplementary Fig. 5d). Compared to the control group, Tg (*mpeg*:GFP), a transgenic line of macrophages, also showed a reduction after *nlrc3* knockdown at 30 hpf (Supplementary Fig. 5e). These results reflected the role of *nlrc3* signaling in primitive hematopoiesis.

Since HSPCs originate from hemogenic endothelial cells in the dorsal aorta[33], mutants or defects in blood vessel or artery genes may lead to hematopoietic defects. To explore this possibility, we examined the vascular endothelial-specific marker *kdrl* and the arterial-specific markers *dlc* at 26 hpf by WISH and found no notable vascular abnormalities in *nlrc3* morphants or mutants (Supplementary Fig. 6a, b). The same trend was observed in Tg (*kdrl*:mCherry) at 28 hpf by confocal microscopy, and no alterations were observed between the different groups (Supplementary Fig. 6d). Furthermore, the expressions of the endothelial and arterial markers *dll4*, *ephrinB2*, *kdrl*, and *deltaC* was not altered relative to the control group, as detected by qPCR (Supplementary Fig. 6c).

To further investigate whether the decrease in HSPCs in *nlrc3* morphants was due to enhanced apoptosis in endothelial cells, we performed TUNEL in Tg (*fli1a*:EGFP) embryos, which labels endothelial cells and were analyzed by confocal microscopy at 28hpf, the results indicated that loss of *nlrc3* caused no increase in apoptotic endothelial cells within the DA (Supplementary Fig. 6e), indicating that the defects in HSPCs in *nlrc3*-deficient embryos were not caused by apoptosis in endothelial cells. Taken together, these data imply that *nlrc3* plays an imperative role in the definitive hematopoiesis of the zebrafish embryos. For the primitive wave, *nlrc3* signaling was indispensable for the creation of neutrophils and macrophages. And this effect is not based on affecting the development of arterial vessels or causing apoptosis. As no previous studies have reported the involvement of *nlrc3* in HSPC generation, we will investigate its underlying mechanism.

### *Nlrc3* regulates HSPC emergence via activation of Notch signaling

To determine the regulatory mechanism by which *nlrc3* signaling regulates HSPCs, we sorted EGFP⁺ cells in Tg (*fli1a*: EGFP) zebrafish embryos at 28 hpf, which was the stage where EHT occurs in the VDA, and these EGFP⁺ cells contained hemogenic endothelial cells (Fig. 4a). Moreover, principal component analysis (PCA) indicated the high quality of our transcriptome profile data (Fig. 4b).

Based on the results of our RNA-Seq assays, a heatmap of differentially expressed genes (DEGs) showed hematologic-associated gene changes in the pathways related to *nlrc3* in the morphant group (Fig. 4c). As well as functional enrichment of common DEGs was

performed, and GO analysis demonstrated changes in the categories of molecular function (MF), cellular component (CC) and biological process (BP) after knockdown *nlrc3* (Fig. 4d). To predict specific downstream pathways, KEGG pathway enrichment analysis was performed, and this analysis revealed that the upregulated DEGs were mainly involved in endocytosis and the MAPK signaling pathway, while the downregulated DEGs were mainly involved in multiple signaling pathways including TGF-β signaling, adherens junction, Wnt signaling, and Hedgehog signaling[34,35]. Among these pathways, Notch signaling was the most enriched in *nlrc3* morphant (Fig. 4e). Notch family genes were downregulated to varying degrees compared with the control groups, the *nlrc3* knockdown groups showed different degrees of downregulation of notch-related genes, especially *notch1a* and *hey1* specifically (Fig. 4f). This trend was consistent with that of hematopoietic-related genes in Gene set enrichment analysis (GSEA) profiles (Fig. 4g, h). These results strongly suggested that Notch signaling might be involved in *nlrc3*-mediated regulation of HSPC emergence.

To further explore this hypothesis, confocal microscopy images of the double-transgenic line Tg (*tp1*: EGFP/*kdrl*:mCherry), which represented active Notch signaling, showed a reduced number of *tp1*⁺*kdrl*⁺ cells in the aortic floor at 30 hpf in *nlrc3*-deficient embryos, and this phenotype was rescued by *nlrc3* mRNA overexpression (Fig. 5a, b). Similar trends were observed in transgenic Tg (*tp1*:EGFP) embryos at 48 hpf (Supplementary Fig. 7a)[36]. Flow cytometry showed that the number of *tp1*⁺ cells was decreased when *nlrc3* was inhibited, and this was restored by *nlrc3* mRNA overexpression (Supplementary Fig. 7b, c). Furthermore, in the RNA sequencing data, *wnt16* showed a significant downregulation, which is considered a major regulator of Notch[37,38]. To further investigate the mechanism of *nlrc3* regulation of Notch, we synthesized the mRNA of *wnt16* for overexpression experiments. The results of WISH showed that the decreased expression of *runx1* at 28hpf due to the loss of *nlrc3* could be partially rescued by overexpression of *wnt16* (Supplementary Fig. 8a, b). The qPCR results also confirmed this phenotype (Supplementary Fig. 8c). On the other hand, as significant regulators of the Notch signaling pathway and hematopoiesis during this specific developmental stage, we have identified genes in the RNA sequencing data that exhibit notable differences, including *vegf* and *evi1*[35,39,40]. In order to further explore the regulatory mechanisms of *nlrc3*, we synthesized their mRNA and performed overexpression experiments on *nlrc3* mutants. However, the qPCR results indicate that the overexpression of *vegfaa*, *vegfab*, and *evi1* cannot effectively rescue the downregulation of *runx1* and *cmyb* caused by *nlrc3* knockout (Supplementary Fig. 9a–c). The results above demonstrate that within the hematopoietic regulatory network governed by *nlrc3*, many genes are perturbed, but it is the Notch signaling that serves as the downstream regulatory role of *nlrc3*.

To provide additional evidence from another perspective, we used the notch-specific inhibitor DAPT (N-[N-(3,5-difluorophenacetyl)-lalanyl]-S-phenylglycine t-butyl ester) to block the Notch signaling pathway to explore the relationship between *nlrc3* and Notch signaling pathway. The WISH results showed that DAPT-treated embryos had decreased frequency of HSPC marker *runx1*, but the overexpression of *nlrc3* could not rescue these effects (Supplementary Fig. 8d). Similarly, overexpression of *nlrc3* could not rescue the levels of DAPT-treated CD41⁺ cells at 72 hpf (Fig. 5c, d). Intriguingly, qPCR results showed that Notch family genes were decreased after DAPT treatment, but overexpression of *nlrc3* mRNA barely rescued this drop, except for *notch1a* and *hey1* at 30 hpf (Fig. 5e), further indicating that *nlrc3* acts in the upstream of Notch signaling. To test our hypothesis by rescue Notch, the double transgenic line Tg (*hsp70*: GAL4/*UAS*: NICD), which contains the Notch intracellular domain (NICD), which can be triggered and enter the nucleus to play key biological roles during Notch activation, was used to ectopically activate Notch signaling by heat-shock

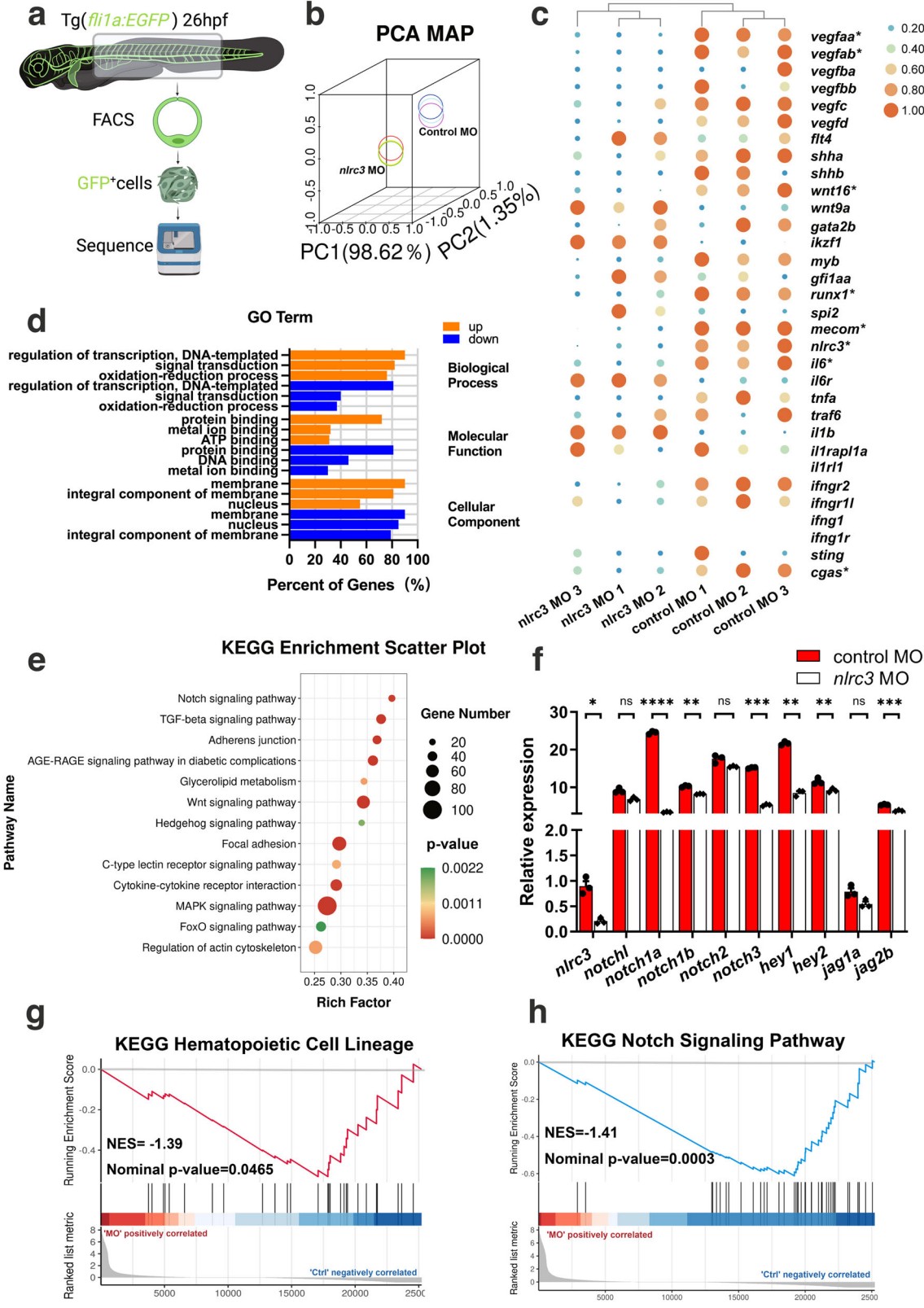

induction after MO injection. In contrast to that of morphants, the overexpression of NICD effectively rescued the expression of *runx1*[+], as shown by WISH (Fig. 5f). Furthermore, induction of NICD rescued the depletion of *cmyb* at 36 hpf and 48 hpf in *nlrc3* morphants (Supplementary Fig. 7d). Altogether, these data supported the hypothesis that *nlrc3* activated the Notch signaling during hematopoiesis in zebrafish.

### *Hey1* acts downstream of *nlrc3*-Notch signaling in HSPC emergence

To explore the mechanism by which *nlrc3* induces Notch activation, qPCR was performed to analyze the expression of Notch ligands and target genes in *nlrc3*-deficient embryos. Consistent with previous results and our RNA-seq analysis, the expression of *notch1a* decreased significantly and this decreasing trend could be rescued by

**Fig. 4 | RNA-seq demonstrates the regulatory mechanism underlying *nlrc3* in HSPC emergence. a** Schematic representation of RNA-seq analysis in this study. **b** Principal-component analysis (PCA) of six sequenced samples shown by the first three principal components (PC1–PC2). Control MO 1, blue; Control MO 2, light blue; Control MO 3, violet; *nlrc3* MO 1, yellow; *nlrc3* MO 2, green; *nlrc3* MO 3, red; **c** Heatmap showing differential expression of hematologic-associated genes. The mark * indicates that there is a significant difference between the control group and morphant group with fold change >2 or fold change <0.5 and *p*-value < 0.05. (4) FDR < 0.05. **d** Representative statistically enriched biological process (BP), molecular function (MF) and cellular component (CC) associated genes of the upregulated and downregulated genes in *nlrc3* morphant group compared with the control group. **e** Representative KEGG enrichment scatter plots of the signaling pathways in the *nlrc3* morphant group compared with the control group. The plots were drawn based on the R version 4.1.3 and ggplot version 3.3.3. The statistical tests were one-sided and adjustments were made for multiple comparisons.

**f** Expression of Notch genes in control embryos and *nlrc3* morphants at 26 hpf by RNA-seq. Error bars, mean ± s.d., ns = 0.0554, 0.0645, 0.1207, *\*P* = 0.0224, \*\**P* = 0.0065, 0.0029, 0.0066, \*\*\**P* = 0.0006, 0.0004, \*\*\*\**P* < 0.0001, the exact *p*-values mentioned above are listed from left to right, by using two-tailed, unpaired Student's *t*-test. *n* = 3 biological replicates. **g** Enrichment plot of the hematopoietic cell lineages between differentially regulated genes in the *nlrc3* morphant group compared with the control group by GSEA (Gene Set Enrichment Analysis). *P* = 0.0465. **h** Enrichment plot of the Notch signaling pathway between differentially regulated genes in the *nlrc3* morphant group compared with the control group by GSEA. *P* = 0.0003. GSEA of *nlrc3* morphant versus control group transcriptional profiles using software GSEA (v4.1.0) and MSigDB, the statistical tests were one-sided and adjustments were made for multiple comparisons. Black bars indicate the individual genes, enrichment is in green. Normalized enrichment score = NES in (**g**, **h**). Source data are provided as a Source Data file.

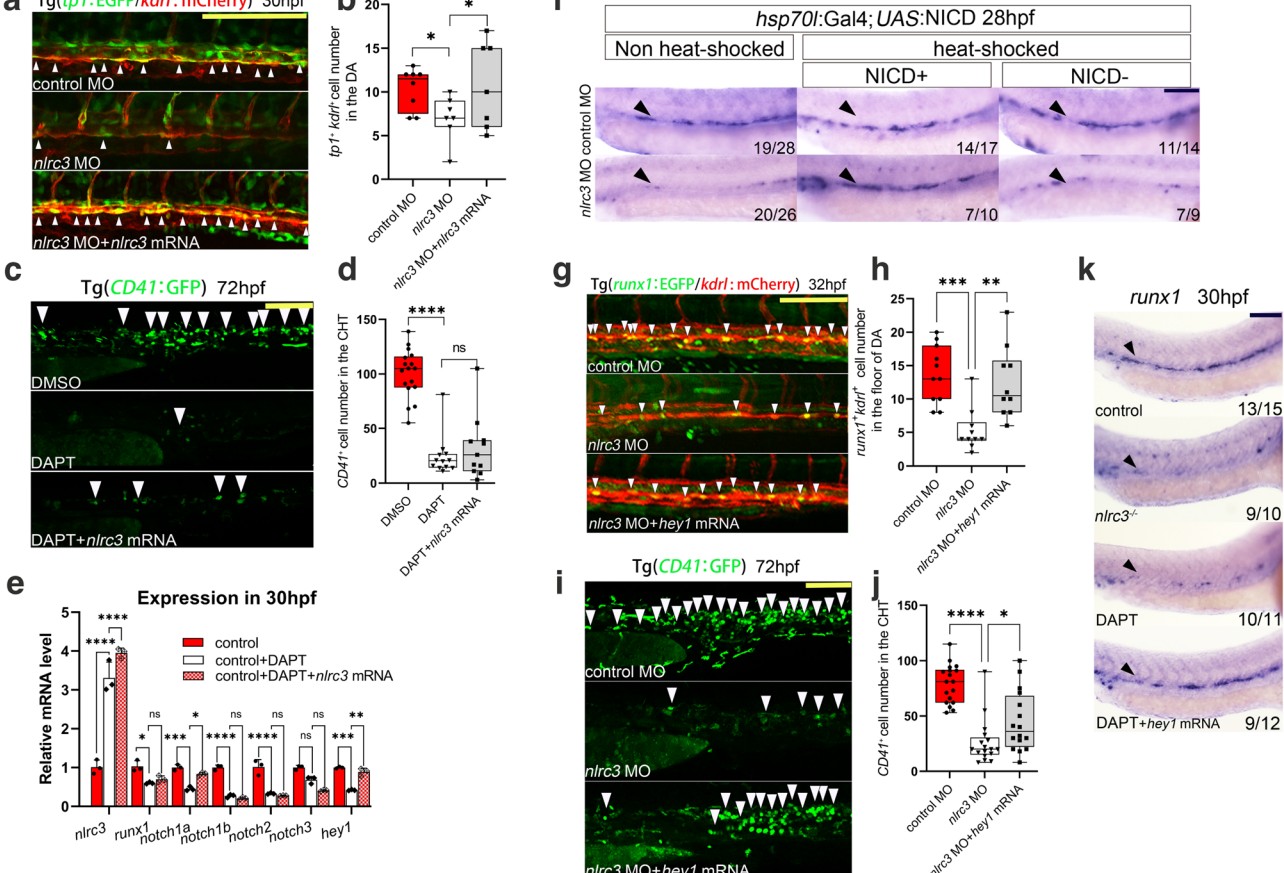

**Fig. 5 | *Nlrc3* regulates HSPC production via activating Notch signaling.**
**a**, **b** Confocal imaging showing Tg (*tp1*:EGFP; *kdrl*:mCherry) embryos at 30 hpf in the floor of the DA (white arrowheads) in control embryos, *nlrc3* morphants, and *nlrc3* morphants with overexpression of *nlrc3* mRNA with quantification (**b**). *\*P* = 0.0394, 0.0221, *n* = 8, 7, 7 embryos. **c**, **d** Confocal imaging showing the number of *CD41*+ cells at 72 hpf in the CHT (white arrowheads) of Tg (*CD41*:GFP) embryos in the control embryos, treatment group with Notch inhibitor DAPT, and treatment group with Notch inhibitor DAPT with overexpression of *nlrc3* mRNA with quantification (**d**). ns = 0.6905, \*\*\*\**P* < 0.0001, *n* = 17, 12, 11 embryos. **e** Expression of *nlrc3*, *runx1*, and NOTCH family genes in the control embryos, treatment group with the Notch inhibitor DAPT, and treatment group with the Notch inhibitor DAPT with overexpression of *nlrc3* mRNA at 30 hpf by qPCR. ns > 0.9999, = 0.2217, 0.4466, *\*P* = 0.0149, 0.0498, \*\**P* = 0.0047, 0.0049, \*\*\**P* = 0.0004, 0.0002, \*\*\*\**P* < 0.0001. **f** *hsp70l*:Gal4; *UAS*:NICD-myc embryos injected with control MO and *nlrc3* MO, WISH for *runx1* was performed at 28 hpf. **g**, **h** Confocal imaging showing the number of *runx1*+*kdrl*+ cells in Tg (*runx1*:EGFP/*kdrl*:mCherry) embryos at 32 hpf in

the AGM (white arrowheads) in control embryos, *nlrc3* morphants, and *nlrc3* morphants with overexpression of *hey1* mRNA with quantification (**h**). \*\**P* = 0.0021, \*\*\**P* = 0.0003, *n* = 11, 10, 10 embryos. **i**, **j** Confocal imaging showing the number of hematopoietic cells in Tg (*CD41*:GFP) embryos at 72 hpf in the CHT (white arrowheads) in control embryos, *nlrc3* morphants, and *nlrc3* morphants with overexpression of *hey1* mRNA with quantification (**j**). *\*P* = 0.0380, \*\*\*\**P* < 0.0001, *n* = 18, 17, 16 embryos. **k** Expression of the HSPC marker *runx1* in control embryos, *nlrc3* mutants, treatment group with Notch inhibitor DAPT and treatment group with Notch inhibitor DAPT with overexpression of *hey1* mRNA at 30 hpf by WISH. Error bars, mean ± s.d., *\*P* < 0.05, \*\**P* < 0.01, \*\*\**P* < 0.001, \*\*\*\**P* < 0.0001, the exact *p*-values mentioned above are listed from left to right, by using one-way ANOVA with –Sidak test in (**b**, **d**, **h**, **j**), two-way ANOVA with Tukey's post hoc test in (**e**), *n* = 3 biological replicates in (**e**). For the box plots in (**b**, **d**, **h**, **j**), box limits extend from the 25th to 75th percentile, while the middle line represents the median. Whiskers extend to the largest value no further than 1.5 times the IQR from each box hinge. Scale bars, 100 μm in (**a**, **c**, **f**, **g**, **i**, **k**). Source data are provided as a Source Data file.

overexpression of *nlrc3* (Supplementary Fig. 8e). Moreover, the expression of *hey1* was significantly downregulated compared with other target genes (Supplementary Fig. 8f, Fig. 4f), suggesting that *nlrc3* plays a regulatory role through *notch1a* and *hey1*.

To prove whether *hey1* was downstream of *nlrc3* signaling, we synthesized *hey1* mRNA for rescue experiments. To explore the effects of *hey1* on the emergence and expansion stages of HSPCs, we used *nlrc3* MOs with *hey1* overexpression to determine the numbers of HSPCs at different time points, and we found that *hey1* could rescue the decreased numbers of *runx1*+*kdrl*+ positive HSPCs at 32 hpf (Fig. 5g, h) and *CD41*+ HSPCs in CHT at 72 hpf (Fig. 5i, j). To validate the relationship between *nlrc3* and *hey1* at the mRNA level, we injected *hey1* mRNA into *nlrc3* mutants to verify its effect on multiple markers by qPCR. The results showed that overexpression of *hey1* had no significant impact on the expression of *nlrc3* but rescued the expression of HSPC markers (*runx1*, *cmyb*) and *hey1*-related genes *edn1* and *plxnd1* (Supplementary Fig. 9h). Furthermore, WISH was performed after the morphant embryos were treated with the Notch signaling pathway inhibitor DAPT, and overexpression of *hey1* mRNA also rescued the decreased expression of *runx1* (Fig. 5k). To examine the ability of HSPC differentiation, we measured the expression of myeloid, erythroid and lymphoid lineages respectively through *Sudan Black*[41], *gata1a*, and *rag1* at 96 hpf and found that *hey1* could rescue these lineages under conditions of Notch inhibitor (Supplementary Fig. 9d–f), indicating that the differentiation of HSPCs was repaired by this pathway. In summary, these findings suggest that *hey1* was downstream of *nlrc3* and that *nlrc3*-Notch-*hey1* axis triggers HSPC emergence and differentiation.

### *Nlrc3* is downstream of the NF-kB signaling pathway in HSPC emergence

Previous reports have revealed that the TLR4−MyD88−NF-kB signaling pathway is involved in regulating the production of HSPCs[1], and correspondingly, *nlrc3* was reported to attenuate Toll-like receptor signaling via modifying NF-kB with adults[7]. Therefore, we continued to explore the connection between NF-kB and *nlrc3* during embryonic hematopoiesis.

First, as previously described, RNA-seq analysis showed that the knockdown of *nlrc3* did not affect the expression of the NF-kB pathway in GSEA (Supplementary Fig. 9i). Specifically, after the knockdown of *nlrc3*, NF-kB-related gene interleukin 1 beta (*il1b*), *nfkbie* and *nfkbiz* were upregulated, *nfkbiab*, *nfkb2* and *nfkbib* were downregulated, *nfkbiaa*, *nfkb1*, *nfkbil1* and *p65* were not statistically difference compared with the control group (Fig. 6a). These results prompted us to examine whether *nlrc3* might be regulated by NF-kB. Second, to explore the relationship between *nlrc3* and NF-kB, IkBaa, the NF-kB inhibitory protein that normally binds to NF-kB and prevents the translocation of NF-kB to the nucleus, was knocked down by MO to increase the NF-kB activity. As shown by qPCR, expression of *runx1* and NF-kB-related gene *il1b* was not significantly increased after overexpression of NF-kB in *nlrc3*−/− mutants compared to the control group (Fig. 6b). To verify this phenotype, we performed WISH and found that, compared to that in the control group, *runx1* was activated due to knockdown of IkBaa. It was not increased in *nlrc3* mutants even with the overexpression of NF-kB (Fig. 6c, d). Besides, JSH-23, a target inhibitor of the nuclear translocation of NF-kB[42], was administered to embryos with *nlrc3* overexpression, and WISH analysis of *runx1* and *cmyb* was performed at 36 hpf and 48 hpf, respectively. Both inhibitors-treated embryos showed significant reduction of *runx1* or *cmyb* expression in the aortic floor, and *nlrc3* mRNA could rescue the phenotypes (Fig. 6e, f, Supplementary Fig. 9g). Similarly, to assess this trend during the expansion of HSPCs, these results were supported by the quantitation of *CD41*+ HSPCs at 72 hpf (Fig. 6g). Third, to further evaluate whether NF-kB induced *nlrc3* activation, qPCR analyses of embryos at 36 hpf showed that *il1b*, the NF-kB subunit *p65*, was

reduced after JSH-23 treated, but the overexpression of *nlrc3* mRNA could not rescue this phenotype. Meanwhile, both expression levels of *nlrc3* and hematopoiesis marker were increased following the enforced expression of *nlrc3* (Fig. 6i). In summary, these data confirmed that *nlrc3* was downstream of NF−kB signaling pathway in transcription and that the NF-kB-*nlrc3*-Notch-*hey1* axis was both necessary and sufficient for HSPC emergence in zebrafish (Supplementary Fig. 9j).

### *Nlrc3* signaling is conserved in the regulation of vertebrate hematopoiesis

To further extend our findings in mammals, we utilized *Nlrc3* knockout mice to explore whether *Nlrc3* signaling was evolutionarily conserved in HSPCs emergence. We first verified the knockout efficiency at the Nucleic acid and protein levels (Supplementary Fig. 10a–c). Then we found *Nlrc3*−/− embryos showed developmental delays and *Nlrc3*−/− FLs are smaller than wild-type at 14.5 days (E14.5) (Supplementary Fig. 10d). Like the phenotype of our zebrafish model, we examined the AGM region of *Nlrc3*−/− embryo to determine whether the production of HSPCs was changed (Fig. 7a). Hematogenic endothelial cells (CD31+ CD41− CD45− TER119−)[43] from the embryonic AGM tissues at E10.5 were sorted and cultured in OP9 for 4 days (Fig. 7b), and then, flow cytometry was performed to assess the proliferative capacity of these HSPCs. c-Kit+CD45+ hematopoietic cells were notably decreased in these *Nlrc3*−/− embryos (Fig. 7c, d). Correspondingly, to obtain more direct phenotypes, flow cytometry was also performed and showed that CD31+Sca-1+CD201+ hematopoietic stem cells were decreased in the AGM of *Nlrc3*−/− embryos at E10.5 (Supplementary Fig. 10e–g).

To verify the phenotype of HSPCs in FL (Fig. 7e), through flow analysis, we found that the proportion and absolute number of LT-HSC populations (Lin− c-Kit+ Sca-1+ CD150+ CD48−) of FL cells from *Nlrc3*−/− embryos were markedly decreased (Fig. 7f, g). To test the function of *Nlrc3*−/− FL-derived hematopoietic stem cells in vitro, by performing a colony-forming unit-cell (CFU-C) assay, we found compromised colony formation abilities of these FL HSCs from *Nlrc3*−/− embryos (Fig. 7h). To verify the functionality of *Nlrc3*−/− embryonic HSCs in vivo, we performed BM transplantation experiments and observed a more than 3-fold reduction in the overall repopulation rate (Fig. 7i, j). We also observed a myeloid lineage bias compared to that in the control group (Fig. 7k). At 16 weeks post transplantation, the proportion of CD45.2 in the BM of recipient mice was reduced by more than 3 times compared to that of wild-type mice (Fig. 7l, m).

To verify the underlying mechanism of impaired HSC function in *Nlrc3*−/− mice, we performed qPCR using mRNA from FL-derived LSK (Lin− c-Kit+ Sca-1+) cells. The results showed that compared with the wild-type group, the Notch family-related genes were downregulated at the transcriptional level, and the expression of *Hey1* was also decreased. This downward trend was consistent with that of *Nlrc3* and the HSC marker *Runx1*, while no significant changes were found in the NF-kB family gene *Il1b* (Fig. 7n). Linking to the reported sequencing data during fetal EHT[28], the expression of *Notch1a* and *Hey1* significantly decreased in HSCs compared to HE, while the expression of *Nlrc3* was increased (Supplementary Fig. 10h). The above results further demonstrated that the NF-kB-*Nlrc3*-Notch-*Hey1* axis orchestrated development of embryonic HSCs was highly conserved in vertebrates (Supplementary Fig. 10j).

## Discussion

Inflammatory signaling is generally believed to be active in response to cell emergencies such as infection. Since inflammatory factors have an essential role in supporting HSPC proliferation and differentiation into mature immune cells in adults under stress-induced hematopoiesis[44,45], the roles of PRRs and their regulatory mechanisms in early embryonic hematopoiesis warrant more attention. In this

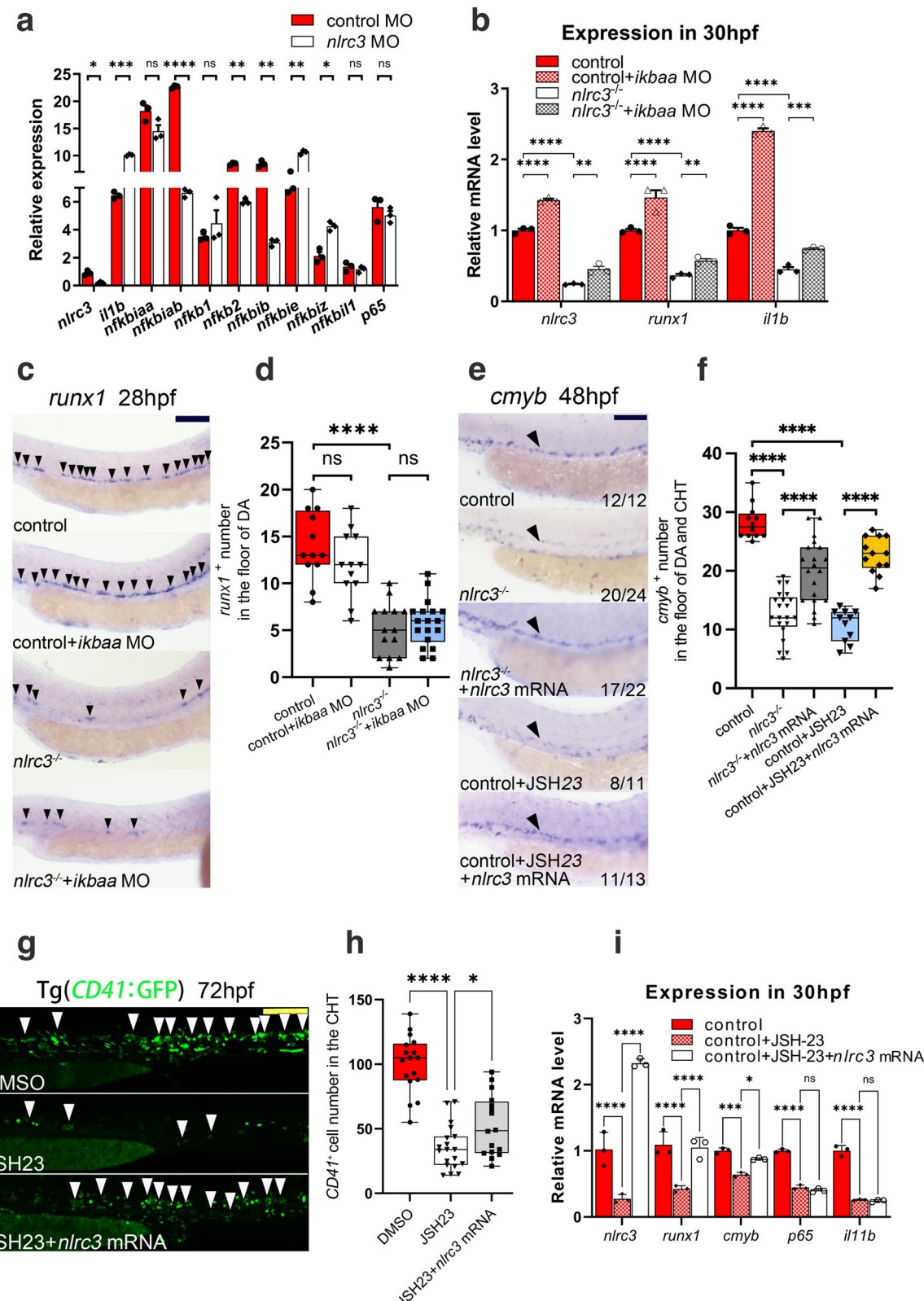

study, we demonstrated that a member of the NLR family, *Nlrc3*, regulates HSPC emergence in vertebrates, which further highlights the roles of inflammasome signaling pathways in hematopoiesis and raises the question of whether their impacts are broader than previously thought. The current question is the role of *NLRC3* in adult blood cells, such as myeloid cells. In the latest reports, *Nlrc3* has been suggested to induce immune tolerance in monocytes and macrophages during

polymicrobial sepsis[46], In this context, the expression of *Nlrc3* in myeloid cells becomes a reference point for monitoring such infections. Combining our analysis of *NLRC3* expression in subpopulations of myeloid cells within the adult bone marrow, we observed enrichment in conjunction with inflammatory factors such as *CSF2R* and *INFGR* (Supplementary Fig. 10i)[47]. On one hand, this indicates that *Nlrc3* plays a crucial role from hematopoiesis to mature immune cells. On the

**Fig. 6 | *Nlrc3* is downstream of the NF-kB signaling pathway for regulating HSPC emergence. a** Expression of NF-kB genes in control embryos and *nlrc3* morphants at 26 hpf by RNA-seq. ns = 0.2176, 0.4498, 0.6039,0.1870, *P = 0.0224, 0.0311, **P = 0.0022, 0.0018, 0.0083, ***P = 0.0002, ****P < 0.0001. **b** Expression of *nlrc3*, *runx1*, and the NF-kB target gene *il1b* in control embryos, *nlrc3* mutants, *nlrc3* mutants with *ikbaa* Mo, and treatment group with *ikbaa* Mo at 30 hpf by qPCR. **P = 0.0041, 0.0053, ***P = 0.0001, ****P < 0.0001. **c, d** Expression of the HSPC marker *runx1* in control embryos, *ikbaa* morphants, *nlrc3* mutants, and *nlrc3* mutants with *ikbaa* morphants at 28 hpf by WISH (black arrowheads) with quantification (**d**) ns = 0.3163, 0.9187, ****P < 0.0001, *n* = 12, 12, 14, 18 embryos. **e, f** Expression of the HSPC marker *cmyb* in control embryos, *nlrc3* mutants, *nlrc3* mutants with overexpression of *nlrc3* mRNA, treatment group with NF-kB inhibitor JSH-23, and treatment group with NF-kB inhibitor JSH-23 and overexpression of *nlrc* mRNA at 48 hpf by WISH (black arrowheads) with quantification (**f**) ****P < 0.0001, *n* = 12, 21, 20, 11, 13 embryos. **g, h** Confocal imaging showing the

number of *CD41*+ cells at 72 hpf in the CHT (white arrowheads) of Tg (*CD41*:GFP) embryos in the control embryos, treatment group with NF-kB inhibitor JSH-23, and treatment group with NF-kB inhibitor JSH-23 and overexpression of *nlrc3* mRNA with quantification (**h**). *P = 0.0387, ****P < 0.0001, *n* = 17, 19, 16 embryos. **i** Expression of HSPC genes and NF-kB genes in control embryos, treatment group with NF-kB inhibitor JSH-23, and treatment group with NF-kB inhibitor JSH-23 with overexpression of *nlrc3* mRNA at 30 hpf by qPCR. ns = 0.8444, 0.9785, *P = 0.0179, ***P = 0.0005, ****P < 0.0001. Error bars, mean ± s.d., *P < 0.05, **P < 0.01, ***P < 0.001, ****P < 0.0001, the exact *p*-values mentioned above are listed from left to right, by using two-tailed, unpaired Student's *t*-test in (**a**), one-way ANOVA with −Sidak test in (**d, f, h**), two-way ANOVA with Tukey's post hoc test in (**b, i**), *n* = 3 biological replicates in (**a, b, i**). For the box plots in (**d, f, h**), box limits extend from the 25th to 75th percentile, while the middle line represents the median. Whiskers extend to the largest value no further than 1.5 times the IQR from each box hinge. Scale bars, 100 μm in (**c, e, g**). Source data are provided as a Source Data file.

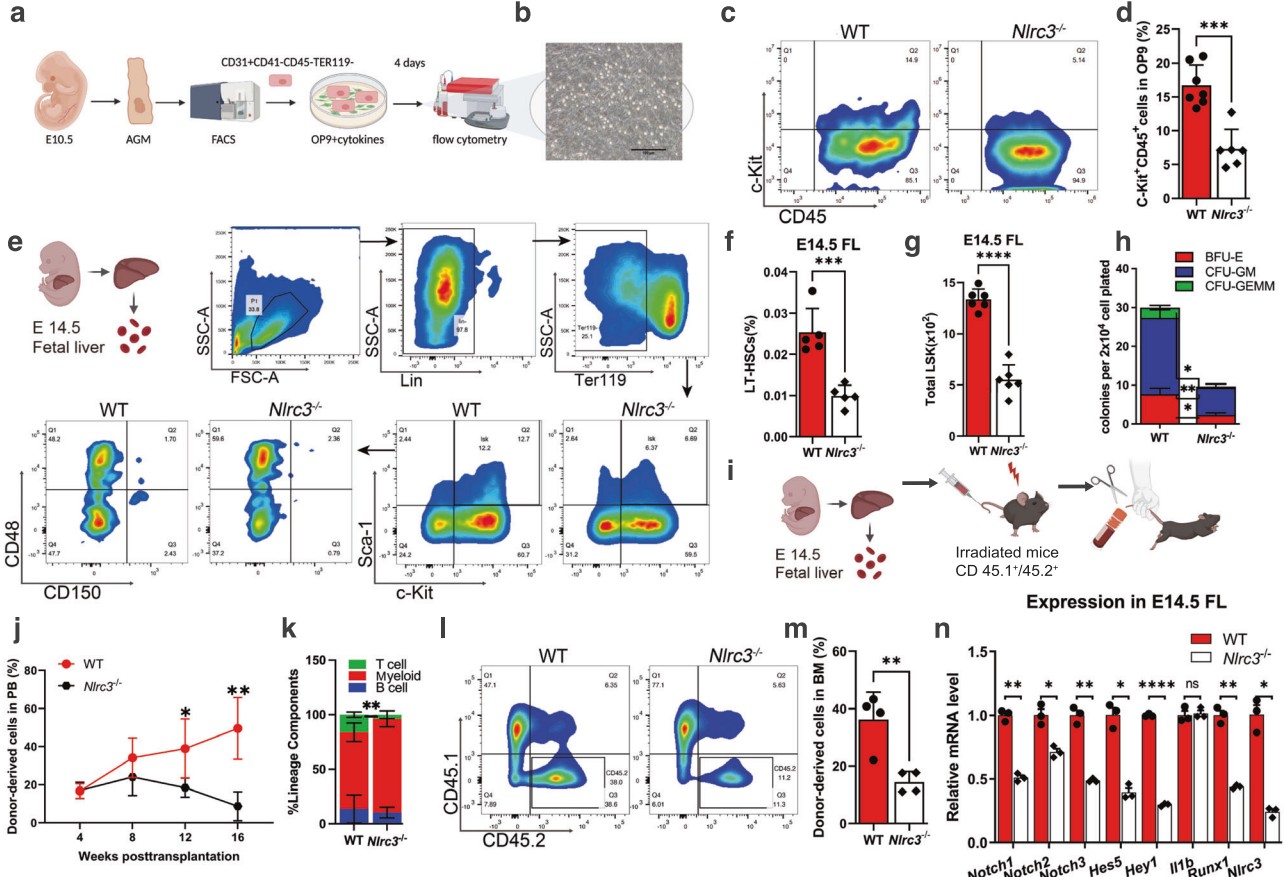

**Fig. 7 | *Nlrc3* signaling is conserved in the regulation of vertebrate hemato-poiesis. a** Schematic representation of AGM-derived hematopoietic endothelial (HE) cells. **b** Representative view of the hematopoietic cells (round) generated from 500 HE cells after co-culture with OP9 stromal cells for 4 days. **c, d** Flow cytometry results showing that after 4 days of OP9 co-culture, the number of c-Kit+CD45+ cells in *Nlrc3* embryos decreased with quantification (**d**). ***P = 0.0001, *n* = 7, 6 embryos. **e** Schematic representation of fetal liver-derived cells performed in this study and flow cytometry results showing that the number of LT-HSC (Lin− c-Kit+ Sca-1+ CD150+ CD48−) cells in *Nlrc3* embryos decreased. **f** Percentage of LT-HSCs in 14.5 embryonic fetal liver of control embryos and *Nlrc3* embryos. ***P = 0.0006, *n* = 5 embryos. **g** Quantification of LSK in 14.5 embryonic fetal liver of control embryos and *Nlrc3* embryos. ****P < 0.0001, *n* = 6 embryos. **h** CFU-C assay of fetal liver showing that the number of CFU-mix, CFU-granulocyte-macrophage (CFU-GM), and CFU-erythroid (CFU-E) was decreased upon knock-down of *Nlrc3*. *P = 0.0114, 0.0132, ***P = 0.0007, *n* = 3 biological replicates. **i** Schematic representation of bone marrow transplantation experiments performed in this study. **j** A total of 5 ×

10⁵ fetal liver-derived cells sorted from WT or *Nlrc3*−/− mice were transplanted with 5 × 10⁵ rescue cells into recipient mice. Peripheral blood (PB) analysis for the percentage of engrafted donor cells at the indicated number of weeks post-transplantation. *P = 0.0473, **P = 0.0038, *n* = 4 biological replicates. **k** At 16 weeks post-transplantation, the PB of recipient mice transplanted with control or *Nlrc3*−/− was analyzed for the percentage of donor-derived B, T, and myeloid cells. **P = 0.0075, *n* = 3 biological replicates. **l** Flow cytometry results show the pro-portion of donor with control or *Nlrc3*−/− in the BM of recipient mice at 16 weeks post-transplantation *n* = 5 biological replicates. **m** Quantification of the proportion of the donor-derived cell in BM. **P = 0.0055, *n* = 4 biological replicates. **n** Expression of *Nlrc3*, *Runx1*, *Il1b*, and Notch-related genes in WT, *Nlrc3*−/− whole FL at E14.5 by qPCR. ns = 0.8168, *P = 0.0137, 0.0205, 0.0107, **P = 0.0078, 0.0039, 0.0063, ****P < 0.0001. Error bars, mean ± s.d., the exact *p*-values mentioned above are listed from left to right by using two-tailed, unpaired Student's *t*-test in (**d, f, g, j, k, m, n**). Illustrations created with BioRender.com. Source data are pro-vided as a Source Data file.

other hand, it suggests that targeting *NLRC3* in specific cells might be a promising candidate therapy.

The contribution of Notch ligands, receptors, and target molecules to hematopoiesis has been extensively elucidated[13,14,48]. It is worth noting that this feature stems from the duality between arteriogenesis and hematopoiesis. Since arteries are thought to be the first niche for HSPCs, the differences in the association between arterial specification and definitive hematopoietic program underscore the intricacy of embryonic HSPCS development and the difficulty of recapitulating it in vitro. Our work not only clarifies the role of the crucial factors upstream of Notch but also expands our understanding of its downstream molecule *hey1*, which has been recognized as a typical Notch downstream gene enriched in the cardiovascular system. Our findings promoted a more complete understanding of Notch and its roles in the regulation of hematopoiesis.

Prior investigations have exhibited that *Wnt16* engages in reciprocal interactions with Notch-related ligands, facilitating the specialization of hematopoietic stem cells[37,49]. Within this study, we observe that silencing of *nlrc3* leads to a notable downregulation of *wnt16*, while the closely linked Notch ligands *deltaC* and *deltaD*, along with *notch3*, remain largely unaltered. This may potentially elucidate that the effects of *nlrc3* signaling on *wnt16* alteration persist until the EHT stage and beyond, encompassing later stages of hematopoietic development. Curiously, the conventional belief is that *wnt16* modulation occurs prior to the emergence of HSPCs, yet it is intriguing that *wnt9a*[50], another member of the Wnt family, which is conjectured to be related to the amplification of HSPCs, exhibits significant upregulation, albeit at a much later juncture around 32 hpf [51]. This outcome extends a broader scope for the exploration of the interplay between *nlrc3* signaling and the Wnt family. What is more, the determination of arterial versus venous fate has been elucidated to hinge upon the Vegf–Notch pathway[52,53]. Interestingly, in the absence of *nlrc3* signaling, varying degrees of downregulation are detected within the Vegf family. Nevertheless, notable downregulation of Vegf receptors *flt4* and *flk1*/*kdrl* remains absent, and the overexpression of *vegfaa* and *vegfab* mRNA fails to effectively rescue the defects in HSPCs. These two adverse outcomes might imply that the *nlrc3* signal, acting as an upstream factor, influences Vegf even before the occurrence of HSPCs, whether it does directly participate in the initiation of HSPCs or not[54]. After all, our understanding remains limited regarding the intricate mechanisms through which Vegf orchestrates the initiation of HSPCs.

The NF-kB family of transcription factors serve as regulators under stress conditions and are regarded as a regulator of the emergence of HSPCs that support hematopoietic homeostasis and regeneration[21]. Previous studies have revealed that *Nlrc3* acts as a negative regulator of NF-KB[6,55], through NF-KB, Myeloid-specific *Nlrc3* deletion induced immunosuppression by improving macrophage glycolysis[46]. In this work, we established the mutual relationship between *nlrc3* and NF-kB in vivo, showing that compared with adults, they have different regulatory relationships in embryonic stages. Different from negatively regulating NF-kB upstream, we propose that *nlrc3* is regulated downstream of NF-kB. Consider the difference between embryonic and adult stages, understanding the evolutionary process of this role switch from embryos to adults requires in-depth research and tracking. Moreover, we delineated the network of the NF-kB pathway in the embryonic emergence of HSPCs, and we provided more perspectives deciphering the complete function of the NF-kB signaling pathway.

Overall, by summarizing the results of the physiological and developmental roles in vivo and in vitro, we demonstrated that the NF-kB-*Nlrc3*-Notch-*Hey1* axis plays a key role in vertebrate HSPC emergence and differentiation. Our findings suggested that functional deficiency in the *Nlrc3* mutant and morphant during HSPCs development under physiological conditions is of major importance since this investigation might be taken into consideration when recapitulation definitive hematopoiesis in vitro.

## Methods

The research complies with all relevant ethical regulations and was approved by the Laboratory Animal Center of Zhejiang University.

### The culture, differentiation, and sorting of embryonic stem cells (ESCs) in vitro

The data of mESCs hematopoietic differentiation and RNA-seq derived from the publications[30]. The hematopoietic differentiation of hESCs was performed using STEMdiff Hematopoietic Kit (STEMCELL Technologies; 05310) following the instructions. Fluorescence-activated cell sorting (FACS) was performed according to the following two schemes, first based on the differentiation time point including day3, day6, day9 and day 12. Second, cells were sorted by the following markers: CD309-Percp/cy5.5 CD31-APC CD34-PE, CD43-APC and CD45-FITC (all kits Biolegend). The total RNA of these FACS-sorted cells was extracted into RNA by using Trizol reagent (Life Technologies; 15596026), and 1 µg RNA was reverse-transcribed into complementary DNA (cDNA) using HiScript II Q RT SuperMix for qPCR (+gDNA wiper) according to the manufacturer's instructions (Vazyme; R223-01).

### Zebrafish and mouse strains

All zebrafish strains were maintained in 28.5 °C system water in accordance with the relevant guidelines of the Laboratory Animal Center of Zhejiang University[17]. Male and female mice from 8 to 12 weeks old were used for all studies. The mice used included C57 wild-type CD45.2, *Nlrc3* knockout mice with CD45.2, and CD45.1 mice. All mice were cultured in a suitable temperature and humidity environment and fed with sufficient water and food (25 °C, suitable humidity (typically 50%), dark/light cycle for 12 h); the morning in which a vaginal plug was observed in the female was defined as embryonic day 0.5 (E0.5). All procedures of the animal experiments were reviewed and approved by the Institutional Animal Care and Use Committee of the Laboratory Animal Center, Zhejiang University (Reference No.: ZJU20230065).

### Injection of morpholino

Splice Modifying morpholinos (MOs) used in our study were ordered from Gene Tools. The MOs were diluted in diethyl pyrocarbonate-treated water at a concentration of 0.2 mM (Standard-MO) and 2 mM (*Nlrc3* MO) with phenol red solution, and 1 nl indicated MO was injected into one-cell stage embryos. The sequences of MOs used in this study are listed in Supplementary Table 1.

### Generation of zebrafish knockout lines using CRISPR/Cas9

In brief, a target site (GGTCTACTGTCGCCCACAGCTGG) in exon 1 of the *nlrc3* gene was chosen, and the guide RNA (gRNA) template was amplified from the pMD-gata5-gRNA scaffold vector. In vitro transcription was performed using 1 µg of template DNA and T7 RNA polymerase. Zebrafish codon-optimized Cas9 plasmid was linearized with *Xba*I, and Cas9-capped mRNA was transcribed using the T7 High Yield RNA Transcription Kit (Vazyme; TR101-01). The size and quality of the capped mRNA and gRNA were confirmed by electrophoresis through a 2% (w/v) agarose gel. Subsequently, 100 pg of gRNA and 400 pg of Cas9 mRNA (New England Biolabs; M0646T) for microinjection into one-cell stage embryos. The following primer pairs and the restriction enzyme *Bst*XI were used to assess the efficiency of genetic disruption: forward5'-ACTTTGGGTCGTCTTGCTTTTTA-3', reverse5'-AACAAATAGGCGAGAACAGCACA-3'. F0 embryos with the highest editing efficiency were raised. Heterozygous F1 fish were identified using DNA sequencing of the offspring of F0 fish outcrosses. The mutant and wild-type animals were obtained from heterozygous crosses.

## Confocal microscopy

Transgenic embryos including Tg(*lcr*:EGFP), Tg(*gata1a*:DsRed), Tg (*mpx*:EGFP), Tg(*lyz*:DsRed), Tg(*mpeg*:EGFP), Tg(*lck*:EGFP), Tg(*fli1a*: EGFP), Tg(*tp1*:EGFP), Tg(*CD41*:GFP) and Tg(*runx1*:EGFP;*kdrl*:mCherry), Tg(*cmyb*:GFP;*kdrl*:mCherry) double-transgenic embryos were anesthetized and mounted on dishes with 0.8% low-melting agarose. Confocal images of the Z sections of the head, DA or CHT region were photographed by an FV1000 microscope (Olympus) to visualize HSPCs and lineages at each time point. The analysis of the images was carried out by Olympus confocal software Image J.

## Whole-mount in situ hybridization

Whole-mount in situ hybridization (WISH) was performed as described[56] with slight modifications. Probes for *nlrc3*, *runx1*, *cmyb*, *gata1a*, *rag1*, *mpx*, *foxn1*, *kdrl*, *dlc*, *l-plastin*, and *mfap4* transcripts were generated using a DIG RNA Labeling Kit (Roche; 11175025910) from relevant linearized plasmids. Embryos were observed and photographed by a Nikon SMZ18 stereo microscope.

## Sudan black staining and *o*-dianisidine staining

Sudan Black staining was used to detect the granules of the granulocytes[57]. Fixed larvae were rinsed in PBT (phosphate-buffered saline, PBS with 0.1% Tween-20; Sigma-Aldrich) twice for 5 min and incubated in 1 mL Sudan Black (Sigma-Aldrich) for 20 min. To stop the staining process, the embryos were washed two times in 70% ethanol for 15 min and passage into 30% ethanol and imaged using a Nikon SMZ18 stereo microscope.

*o*-Dianisidine staining was used to check the hemoglobin; a brown color indicated the presence of hemoglobin in zebrafish embryos. Before staining, a stock solution of 100 mg o-dianisidine dissolved in 70 mL ethanol was prepared at 4 °C in the dark. For staining, live embryos were exposed to an o-dianisidine working solution consisting of 2 mL o-dianisidine stock solution, 0.1 m sodium acetate (pH 4.5), and 0.65% H2O2 in a small glass bottle for 3–5 min at room temperature. To stop the staining process, the embryos were washed two times in PBS and were imaged using a Nikon SMZ18 stereo microscope.

## Quantitative real-time PCR analysis

Total RNA was isolated from FACS-sorted cells or each time point dissected regions that included caudal hematopoietic tissue (CHT) or heads of zebrafish by using Trizol and 1 μg RNA was reverse-transcribed into complementary DNA (cDNA) using HiScript II Q RT SuperMix for qPCR (Vazyme; R222-01) according to the manufacturer's protocol. Quantitative PCR (qPCR) was performed on a Bio-Rad CFX96 system using ChamQ Universal SYBR qPCR Master Mix (Vazyme; Q711-02/03) and samples were run in duplicate or triplicate with R3 biological replicate pools/condition. Each sample was performed in triplicate and all results were normalized to the expression of Actin. Data are represented as mean 6 standard deviations (SD), and significance was determined by two-tailed Student's *t*-tests between control and experimental groups. The PCR primers used are listed in Supplementary Table 2.

## mRNA synthesis and injections

For messenger RNA (mRNA) synthesis, total RNA was extracted from zebrafish wild-type embryos and reverse transcribed into cDNA. Specific primers were utilized to amplify the opening reading frame of *nlrc3*, *hey1*, *wnt16*, *gata2b*, *vegfaa*, *vegfab*, *evil*. The PCR product was cloned into the pCS2+ plasmid and validated by bidirectional sequencing. The recombined pCS2+-*nlrc3*, *vegfaa*, *vegfab*, *wnt16*, *evil* plasmid was linearized by NotI and pCS2+- *hey1*, *gata2b*, plasmid was linearized by BamHI and purified by FastPure Gel DNA Extraction Mini Kit (Vazyme; DC301-01). All of these capped full-length zebrafish mRNAs were generated by using SP6 mMESSAGE mMACHINE mRNA transcription synthesis kit (Invitrogen; AM1344) and cleaned up using

RNA clean Kit (TIANGEN; DP412) according to the manufacturer's instructions. Then, 100 pg purified *nlrc3* and *hey1* mRNA was injected into one-cell stage embryos alone or in combination with specific MOs.

## TUNEL assay and staining

The TUNEL assay was performed using the fluorescein-based In Situ Cell Death Detection Kit (Roche; 12156792910) in accordance with the manufacturer's instructions. Manually dechorionated Tg(*fli1a*:EGFP) transgenic embryos were fixed in 4% PFA at 4 °C overnight, washed three times with PBST and dehydrated in 100% methanol at −20 °C for more than 2 h. After gradual rehydration, embryos were washed with PBST three times, followed by proteinase K and acetone treatment. After washing 3 times with PBST, the permeabilized embryos were incubated in a mixture containing labeling solution and enzyme solution at a ratio of 9:1 at 4 °C overnight. Finally, the embryos were washed three times with PBST and then were captured by confocal microscopy. Note that anti-GFP (Invitrogen; A-11120) and Alexa Fluor 488-conjugated goat anti-mouse (Invitrogen; A-10680) antibodies were used as the primary and secondary antibodies, respectively.

## RNA-seq and bioinformatics analysis

RNA-seq analysis was performed by LC-Bio Technology Co., Ltd[4]. Total RNA was extracted from sorted EGFP+ cells in Tg(*fli1a*: EGFP) zebrafish embryos by using TRIzol Reagent following the manufacturer's instructions. cDNA libraries were constructed by SuperScript™ II Reverse Transcriptase (Invitrogen; 18064014), and 150-bp single-end reads were generated on an Illumina HiSeq 6000 platform following the vendor's recommended protocol. After removing the low-quality bases and undetermined bases by Cutadapt software (version: cutadapt-1.9), we used HISAT2 software (version: hisat2-2.0.4) to map reads to the genome (Ensembl Danio reio v96). The mapped reads of each sample were assembled using StringTie (version: stringtie-1.3.4d.) with default parameters. Then, all transcriptomes from all samples were merged to reconstruct a comprehensive transcriptome by using gffcompare software (version: gffcompare-0.9.8.). After the final transcriptome was generated, StringTie and ballgown were used to estimate the expression levels of all transcripts and perform expression level for mRNAs by calculating FPKM. The differentially expressed mRNAs were selected with fold change >2 or fold change <0.5 and *p*-value < 0.05 by R package edgeR (http://www.r-project.org/) or DESeq2, and then analysis GO enrichment and KEGG enrichment to the differentially expressed mRNAs.

## Chemical treatment

Zebrafish embryos were exposed to chemicals from the 10-somite stage to 24 hpf or 36 hpf. For evaluation of HSPCs development, exposure ranged until fixation and was subjected to WISH for *runx1*, *cmyb* and *rag1*. Siblings treated with 0.1% DMSO in Hotter buffer were taken as controls. Embryos with 100 μM DAPT (Calbiochem; 565770)[17] or 300 mM JSH-23 (Selleck; S7351)[1] were treated as treatment groups.

## Heat-shock treatment

For induction of *hsp70l*: Gal4-driven NICD overexpression, embryos were placed to 37 °C heat shock at 18 hpf for 50 min. The embryos were then kept at 28 °C until each time point when they were fixed and processed for WISH.

## Flow cytometry analysis and cell sorting

Cells derived from in vitro human ESC hematopoietic differentiation or tissue of mouse were harvested and suspended in PBS with 2% fetal bovine serum (FBS). Before antibody incubation, cells were blocked with an anti-CD16/32 antibody (eBioscience). Fluorescence-activated cell sorting (FACS) was based on the following markers: TER119- APC/Cyanine7, CD31-APC, CD45-FITC, CD41-PE/cy7, c-Kit-PE/ cy5. Sca-1-APC, CD150-FITC, CD48-PE, CD201-FITC, CD45.1-AF-700

(all kits Biolegend). A different lineage cocktail (Lin) was used, including CD3, CD4, CD8, Gr1, B220, IgM and Ter119[58,59]. DAPI was included to omit dead cells. Cells were stained in PBS/2% FBS for 30 min at room temperature and sorted out using BD AriaII (BD Biosciences).

As for zebrafish, embryos were manually dechorionated with pronase, anesthetized in tricaine, dissociated with liberase, and triturated using a P1000 pipette. The resulting single-cell suspension was filtered with a 40-μm cell strainer and resuspended in PBS. Flow cytometric acquisitions or FACS were performed on BD Fortessa and BD AriaII (BD Biosciences), respectively. Graphs were prepared in FlowJo.

### Mouse embryo dissociation and FACS

E10.5 AGMs (36–40 somite pairs) or E14.5 fetal livers were isolated from 6- to 8-week-old pregnant female mice, the tissues were isolated under a dissecting microscope, and cell suspensions were prepared in PBS containing 2% FBS by repeatedly flushing through needles ranging from 18- to 27-gauge. The cells were then passed through a nylon mesh with a pore size of 70 μm, and red blood cells were lysed using RBC lysis buffer (eBioscience; 00-4333-57). After dissociated by collagenase from WT and *Nlrc3*−/− embryos, at least 500 CD31+CD41−CD45−TER119− cells from E10.5 AGMs were sorted by FACS and cultured on mouse OP9 stromal cells and supplemented with hematopoietic cytokines (50 ng/mL stem cell factor, 50 ng/mL IL3, 20 ng/mL FLT3 ligand). After being cultured for 4 days, semi-adherent cells were carefully harvested for flow cytometry. For staining LT-HSCs, the E14.5 fetal liver cells were first incubated with Fc-Block followed by biotin-conjugated lineage marker antibodies (PE/cy5-conjugated c-Kit, APC-conjugated Sca-1, FITC-conjugated CD150, PE-conjugated CD48). All antibodies were purchased from eBioscience or BioLegend. FACS data were analyzed using FlowJo software.

### In vitro methylcellulose colony-forming assays

Fetal liver suspensions were prepared in Iscove's Modified Dulbecco's Medium (IMDM) supplemented with 2% FBS, and red blood cells were lysed using an RBC lysis buffer. A total of $3 \times 10^4$ FL cells were seeded in MethoCult M3434 (STEMCELL Technologies, 03434) containing recombinant mouse stem cell factor (SCF; 50 ng mL−1), recombinant mouse IL-3 (10 ng mL−1), recombinant human IL-6 (10 ng mL−1), and erythropoietin (U mL−1). All assays were performed in triplicate in a 35-mm culture dish incubated at 37 °C in humidified air containing 5% $CO_2$ for 12 days. The colonies were counted based on standard morphological criteria. BFU-E (burst-forming unit-erythroid), CFU-GM (colony-forming unit-granulocyte/macrophage), and CFU-GEMM (colony-forming unit-granulocyte/erythroid/macrophage/megakaryocyte) were classified and enumerated based on morphological recognition.

### Transplantation experiments

Competitive reconstitution assays were performed as previously reported[6,7]. Eight-week-old CD45.1 mousse was lethally irradiated to a split dose of 9 Gy X-ray. Embryo equivalent cells from WT (CD45.2) or *Nlrc3*−/− (CD45.2) fetal liver were mixed in a 1:1 and 1:4 ratio with $1 \times 10^6$ CD45.1 bone marrow cells and injected into recipient mice into the tail vein. Then peripheral blood was collected from the tail and stained for CD45.1 and CD45.2 antibody at indicated time points. At 16 weeks after transplantation, bone marrow cells and peripheral blood were obtained and detected by flow cytometry.

For competitive transplantation, 8-week-old CD45.1+ mice were lethally irradiated to a split dose of 9 Gy X-ray. Nucleated donor cells (CD45.2+) obtained from E14.5 WT and *Nlrc3*−/− fetal livers were mixed with freshly isolated bone marrow nucleated cells (CD45.1+) in a 1:1 and 1:4 ratio, and the mixture was injected into the caudal vein of CD45.1+ mice. Then peripheral blood was collected from the tail and stained for CD45.1-PE and CD45.2-APC antibody at indicated time points from the recipients 4, 8, and 12 weeks for flow cytometric analysis. At 16 weeks after transplantation, bone marrow cells and peripheral blood were obtained and detected by flow cytometry.

### Statistical analysis

All statistical analyses were generated using GraphPad Prism software (Prism 9). Values are shown as the mean ± standard deviation (SD). The sample size for each statistical analysis is provided in the figure legends. Where indicated, two-tailed, unpaired Student's *t*-test was used for comparisons between two groups, whereas one-way ANOVA was used for more than two groups' comparisons. The log-rank test was used to analyze the survival curves. *$P < 0.05$ was considered statistically significant, **$P < 0.01$, ***$P < 0.001$, ****$P < 0.0001$; n.s., not significant.

### Reporting summary

Further information on research design is available in the Nature Portfolio Reporting Summary linked to this article.

### Data availability

Source data is available for all main figures and supplementary figures in the associated source data file. The RNA-seq data generated in this study have been deposited in the Gene Expression Omnibus (GEO) under the accession number GSE248871. All the relevant data supporting this study are available from the corresponding author upon request. Source data are provided with this paper.

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

## Acknowledgements

We thank Feng Liu and Wenqing Zhang for the generous gift of zebrafish strains, Fudi Wang, Yan Li, Pengfei Xu, Zhipeng Ma, Xuyan He and Paul Liu for help and guidance. This work was supported by grants from the National Key Research and Development Program of China (2022YFA1103500) to P.X.Q. the National Natural Science Foundation of China (82130003 to H.H.; 82200255 to M.Z.; 82222003, 92268117, 82161138028 to P.X.Q., 82100123 to S.F.W.; 82200233 to H.H.L.). the Key R&D project of Zhejiang Provincial Science and Technology Department (2021C03010) and the Leading Innovative and Entrepreneur Team Introduction Program of Zhejiang (2020R01006) to H.H. Program of Shanghai Academic Research Leader (20XD1403500), Shanghai Science Technology and Innovation Action Plan (21Y31920400), Clinical Science and Technology Innovation Project of Shanghai Shenkang Hospital Development Center (SHDC12020128) to J.H.L. Postdoctoral Research Foundation of Zhejiang Province in China (ZJ2021125) to R.X.T. We thank Yinniang Li, Xinghui Song and Xiaoli Hong from core facility platform of Zhejiang University School of Medicine and laboratory animal center of Zhejiang University for their technical support.

## Author contributions

H.H. conceived the project. H.H., S.C., H.L., R.T. and P.Q. made contributions to design the experiments. S.C., H.L. and R.T. performed most of the zebrafish experiments. S.C., R.T. and P.Q. wrote the manuscript. W.S. performed the experiments in mouse. Q.L. helped to generate the plasmid constructs and hESCs experiments. S.W. performed the co-occurrence clustering analysis and flow sorting. C.F. performed computational analysis and bioinformatics analysis. H.C., X.L., M.Z. and Y.X. validated the HSPCs phenotypes and provided useful insights and reagents. M.C. and J.L. made contributions during the article revision process. Conceptualization, reviewing and supervision: J.L., P.Q., H.H. All authors have read and approved this manuscript.

## Competing interests

The authors declare no competing interests.
