## [Peer Review File · Nature Communications]

Nlrc3 signaling is indispensable for hematopoietic stem cell emergence via Notch signaling in vertebratesREVIEWER COMMENTS

Reviewer #1 (Remarks to the Author):

In this manuscript, the authors described the regulatory role of *Nlr3* during embryonic hematopoiesis in vertebrates. According to analysis of the published scRNAseq data in zebrafish CHT, mouse AGM and human AGM, they showed that *nlr3* is expressed both in ECs and HSPCs. Additionally, they detected *NLRC3* expression in HE and HSPCs induced from mESC and hiPSCs, although they didn't validate the features of induced-HE or -HSPCs. Then, they found HSPC production and differentiation were impaired in *nlr3* KD/KO zebrafish embryos. By RNA-seq and functional assays, the authors verified that NF-kb and Notch signaling (through *hey1*) pathways stand upstream and downstream of *nlr3*, respectively. Finally, using *Nlr3* genetic mutant mouse model, they showed that the number and function of HSPCs in AGM and fetal liver were significantly decreased, suggesting a conserved role of *Nlr3* in mice.

Major concerns:

1. My main concern is that what the cell-specific role of *nlr3* is. The expression of *nlr3* seems broad in many cell types, including ECs and HSPCs in zebrafish CHT, and HEC, EC and HSPC in mouse and human AGM. Given that the main finding of the manuscript, i.e., *nlr3* is a regulator for HSPC emergence in zebrafish, I wondered what is the expression pattern of *nlr3* in the AGM region in zebrafish, but not in the zebrafish CHT as shown in Figure 1 A by the authors, and what's the EC, HEC or HSPC specific regulatory role of it?
2. Since HSC emergence is fine-tuned and controlled by inflammatory signaling, not only NF-kb and notch signaling, but also by the primitive myeloid cell and their secreted cytokines. For example, primitive neutrophils secrete *Tnfa* to activate NF-kb and notch signaling to initiate HSC generation (PMID: 25416946); primitive macrophages also dynamically interact with emerging hematopoietic cells and enhance HSC generation etc (PMID: 25686881). In this manuscript, the authors characterized that the primitive neutrophils and macrophages were also reduced in *nlr3*-deficient embryos, would it be possible that the observed HSPC generation defect was a secondary effect caused by the reduced primitive cells? What's the relationship of primitive immune cells with these HEC/HSC defect?
3. The similar regulatory mechanism of *nlr3* in regulating HSC generation and differentiation was not convincing as they described in Supplemental Figure 8C and was not fully supported with sufficient data, see page 16, 1st paragraph.
4. Moreover, since *nlr3* was expressed in both EC and HSPC in the CHT as the Figure 1A shown above, how to rule out the cell extrinsic regulation on HSPC differentiation in the CHT, for example, from the CHT ECs pockets?
5. *Nlr3* conditional KO or overexpression approaches specifically in ECs or hematopoietic cells, should be considered for determining the exactly regulatory role of *Nlr3* during HSPC development.
6. Since the Notch signaling is dynamic from artery/vein specification to HEC/HSPC determination, its activity timing should be considered in the rescue experiments.

Minor concerns:

1. The title of Figure 1 was not consistent with the data in the figure, and was exactly the same to sFigure1.

Reviewer #2 (Remarks to the Author):

The manuscript titled “Nlrc3 signaling is indispensable for hematopoietic stem cell emergence via Notch signal in vertebrates” describes a comparative approach for addressing factors governing HSPC emergence and differentiation in zebrafish and mice. The authors leveraged the utility of the zebrafish model to address the role(s) of a specific pathogen recognition receptor gene for potentially activating Notch governed pathways leading to emergence of myeloid and erythroid lineage development. By using combinations of *nlrc3* mutants and morphants, whole mount in situ hybridization, gene expression analyses coupled with external scRNAseq data, the authors demonstrated a unique role for NLRC3 for HSC and PC development in zebrafish. They then used this information to see if *nlrc3* had an evolutionarily conserved role for HSPCs. Using appropriate murine models coupled with RNAseq, flow cytometry and bone marrow transplantation, the authors provide compelling evidence that the role of *nlrc3* for HSPCs is conserved for vertebrates and thus presents a novel finding. This reviewer had some difficulty following the developing story about Nlrc3 owing to the data-dense results section that lacked proper definition of genes and associated markers used for delineating the role of Nlrc3. This is important as the manuscript should be accessible to a wide readership. Below the authors will find comments and suggestions that are meant to improve the overall accessibility of the results for broad readership.

Major comment: The study utilized a variety of different transgenic zebrafish for demarking different primitive and definitive cell types but rarely were there proper definitions/descriptions for the specific transgenics so that a reader from a related field of study could follow the emerging story without having to look them up. For example, use of the *kdr1* transgenic fish was first mentioned on page 6 but it was not until page 11 that the function of *kdr1* was described—e.g. that *kdr1* is the receptor for VEGF. Reader should not have to go to supplemental section to know what these markers represent for HSPC development. A table describing all transgenics used in this study would be useful.

General comment about the results section: It was not always clear if the authors were discussing results from morphants or mutant zebrafish. The authors used morpholinos for the Tg fish, and they should clearly state this. Very little discussion follows the results section and therefore the manuscript could benefit from a combined results/discussion style with a concluding statement at the end.

Minor:

For the purposes of review, it would have been very helpful to have numbered lines and pages.

The zebrafish genome is known to encode multiple Nlrc genes—can the authors mention how many if zebrafish code for Nlrc3 paralogs and that they Nlrc3 gene studied in this manuscript is orthologous to human Nlrc3.

Page 2, line 20: citations 10-12 are not for teleosts

Page 3, lines 1-3: There is a reference for Hey2 but not for Hey1 nor is there any description what Hey gene products do—GeneCards indicates that these are helix-loop-helix transcription factor—that would be good to know.

Page 3: line 8. Nlrc3 instead of Nlcr3.

Page 3, line 14: authors should cite the “including published data”

Page 3: Line 16: “Mechanistically...Hey1”. Qiao et al 2012 and Liu et al. 2017 also

suggested that *Nlr3* might be regulating actions of *NfκB*.

Page 4: EHT was defined in the abstract, but it should be redefined in the main text.

Page 4: line 10: “And the expression...HSPC development”. *Nlr3* and *Tlr4* expression have similar expression levels in mice, but it does not seem to hold true for zf and humans. Figure 1 indicates that zf *Nlr6* is expressed at relatively high levels but not mentioned—is there any information on potential roles of *Nlr6* for HSPCs? On a similar topic, NLRs interact with ASC and Caspase to form inflammasomes when acting as cytosolic PRRs but nothing is said about ASC or CaspA/B expression during HSPC development or how the inflammasome itself could influence lineage development. Do the authors suspect that a similar phenotype would be observed for zebrafish *Asc*^{-/-} fish (Li et al. 2020) as that for the *Nlr3* mutant and morphants?

Question: Could there be a role for STING in regulating NLRC3 during different stages through HSPC development? Just curious owing to the involvement of STING during NLRC3/NFκB mediated responses.

Figure 1A: There are two genes for zf TNF and IFNG—paralogs should be included especially since their expression is associated with Notch regulating HSPCs.

EHT defined in abstract but not when it appears in the main manuscript.

Page 5: “was highly correlated”, Sup figure 1B. This implies regression analysis, but I did not see that in the supplemental figure or in methods. Was regression analysis a part of the bubble plot and hierarchical clustering?

The authors conducted a very thorough study that implicates *Nlr3* as a contributor for HSPC development and proposed a signaling/regulatory cascade—the manuscript might benefit from a graphical description of the proposed “NFκB-NLRC3-NOTCH-HEY1” signaling pathway for the reader so that they can better appreciate the pivotal role of NLRC3.

NICD needs to be described—readers from other disciplines would have no idea what this represents.

Figure 4B, three dimensional PCA is hard to appreciate—PC3 (0.02%, very low) could be eliminated from the plot

Figure 4 legend: The authors need to adjust their A-F lettering. There is no F.

Reviewer #3 (Remarks to the Author):

The manuscript investigates the Oligomerisation (NOD) like receptor 3 in endothelial-to-hematopoietic transition (EHT) and HSPC emergence. EHT and HSPC emergence are known to be pivotally regulated by the Notch pathway, which in this context is induced amongst other by inflammatory signals and NF-κB. NLRC3 was previously shown to express in various immune cells and its modulation to affect proinflammatory cytokines.

Here, the authors suggest *nlr3* as an upstream activator of Notch activity and the Notch downstream *hey1* gene, and NF-κB signaling as an upstream inducer of *nlr3* transcriptional

activity.

Various expression analyses in three species (zebrafish, mouse and human) relying amongst other on published data and on own data from mouse/human differentiating embryonic stem cells, indicate that *nlrc3* is expressed during development in endothelial and early hematopoietic cells. Elaborate genetic and rescue experiments performed in zebrafish convincingly position *nlrc3* as an upstream regulator of Notch activity required for HSPC emergence. *Nlrc3* appears to mediate its effects on the Notch pathway especially by upregulating *notch1a* and *hey1* expression, which when overexpressed could partially compensate the loss of *nlrc3* during definitive HSPCs production. In the primitive wave of hematopoiesis, loss of *Nlrc3* affected primitive myeloid but not primitive erythroid cells, indicating that it perhaps plays a separate (yet unexplored) role in the myeloid lineage. Importantly, inhibition of NF- κ B affected *Nlrc3* expression, but *nlrc3* expression could not rescue inhibition of NF- κ B, thereby positioning NF- κ B upstream of *nlrc3*.

While the manuscript convincingly places *nlrc3* “between” NF- κ B and Notch by various experiments performed in three species and different experimental models, it does not address in any way and in any model the interaction with other major regulators of Notch and hematopoiesis at this precise stage of development such as *vegf*, *evi1*, *shh*, *wnt16*, *gata2b* (see also following manuscripts: PMID: 27638855, PMID: 12110173, PMID: 21654806, PMID: 15737934, PMID: 27015586 and others).

Major comments:

1.) See also major comment above. Is the expression of *vegf*, *evi1*, *shh* affected by loss of *nlrc3*, and respectively does loss of *nlrc3* affect the expression of *veg*, *evi1* or *shh*? Are *vegf*, *evi1*, *shh* required for *nlrc3* to mediate its effects on HSPC development? Can *notch* be induced by *vegf* or *evi1* in the absence of *nlrc3* and can such induction compensate for the loss of *nlrc3*? Do these pathways interact, are they in parallel required to induce HSPCs from endothelial cells?

Given the multitude of data produced around the NF- κ B => *Nlrc3* => Notch axis, it is hard to understand why these other major regulators of exactly this step are completely experimentally ignored, and not even adequately discussed.

2.) How is the expression of *gata2b* affected by loss of *nlrc3* or *nlrc3* overexpression? Can *gata2b* overexpression rescue hematopoiesis in *nlrc3* morphants or mutants?

3.) Notch regulates EHT, but at which stage and in which cell type does *nlrc3* precisely induce its effects? This is not fully elucidated, and some results are presented in a confusing manner.

Figure 1 shows analysis of published data bases across three species indicating that *nlrc3* is expressed in endothelial and early hematopoietic cells. Expression is found in both venous and arterial endothelial cells, it is unclear from these data at what stage *nlrc3* may regulate HSC emergence. What is the role of *nlrc3* in venous cells, is there potentially a separate role than the here delineated role in HSPC development,

Fig. 1A: shows *nlrc3* expression in cells from the caudal hematopoietic tissue. However, HSCs are formed from specialized endothelial cells (ECs) in the ventral wall of the dorsal aorta (VDA), within the AGM region. These early blood stem cells subsequently migrate to the caudal hematopoietic tissue, which supports HSC expansion, before HSCs finally home to the kidney marrow. Thus, *nlrc3* expression in cells of the CHT would mark expression in cells in which hematopoietic fate has already been established. This is not as helpful for the here proposed role of *nlrc3* in HSPC emergence, please comment.

Fig. 1B. Mouse cells are compared (Aorta Gonad Mesonephros (AGM)-derived HSPCs, fetal liver (FL)-derived HSPCs, venous endothelial cells (VECs), arterial endothelial cells (AECs), pre-hemogenic endothelial cells (pre-HECs) and hemogenic endothelial cells

(HECs). Here the authors conclude that “the expression of *Nlr3* increases with hematopoietic maturation, especially when comparing HECs with HSPCs, which is the stage that EHT occurs.” When looking at the results, it remains however again unclear at which stage exactly *nlr3* should exert its activity. Puzzling is for example its expression in arterial as well as venous endothelial cells (AECs, VECs) in the same plot. Fig. 1C: shows expression in all analyzed human cell types. Supplemental Figure 1 further compares expression of *Nlr3* in fetal liver HSPCs, bone marrow HSPCs and in cells from differentiating mouse and human embryonic stem cells. A parallel is drawn between *Nlr3* and *Tlr4* and *IFN- γ* expression, since these genes are also involved in HSC emergence.

Overall, these data primarily show that *Nlr3* is expressed in endothelial and hematopoietic cells and that its expression associates with that of other inflammation associated genes. It is unclear in fact, how ubiquitous the expressed of these genes is? It would be more informative to show association/clustering/co-expression with lineage-specific markers, such as markers of endothelial and especially hematopoietic identity previously implicated in EHT (*gata2*, *runx1*, *cmyb*, *evi1*). These analyses need to be shown.

Minor comments:

- 1.) Suppl. Fig. 6: what are the top differentially regulated genes?
- 2.) Suppl. 7 title: please correct typo

Responses to the Reviewers' Comments

We greatly appreciate the time and effort of reviewers in providing constructive critiques of our initial manuscript. Their valuable comments and suggested experiments help very much to improve the quality of our manuscript. Based on their constructive comments, especially the questions about the effects of *Nlrc3* on known hematopoiesis regulators, as well as the timing and lineage specificity of *Nlrc3* action, we have clarified the experimental details and performed a series of critical experiments to improve the quality of our manuscript, such as exploration and analysis of the biological informatics data, rescue experiments based on known hematopoiesis regulators, as well as in situ expression experiments of *Nlrc3*. The detailed point-by-point responses to reviewers' comments are shown as below.

REVIEWER COMMENTS

Reviewer #1 (Remarks to the Author):

In this manuscript, the authors described the regulatory role of Nlrc3 during embryonic hematopoiesis in vertebrates. According to analysis of the published scRNAseq data in zebrafish CHT, mouse AGM and human AGM, they showed that nlrc3 is expressed both in ECs and HSPCs. Additionally, they detected NLRC3 expression in HE and HSPCs induced from mESC and hiPSCs, although they didn't validate the features of induced-HE or -HSPCs. Then, they found HSPC production and differentiation were impaired in nlrc3 KD/KO zebrafish embryos. By RNA-seq and functional assays, the authors verified that NF-kb and Notch signaling (through hey1) pathways stand upstream and downstream of nlrc3, respectively. Finally, using Nlrc3 genetic mutant mouse model, they showed that the number and function of HSPCs in AGM and fetal liver were significantly decreased, suggesting a conserved role of Nlrc3 in mice.

Answer: We thank the reviewer's general remarks on our study. It must be clarified that the data we used for mESCs came from previously published sequencing data.¹ The key point is that, to be accurate, we used hESCs (human Embryonic Stem Cells), rather than hiPSCs, which were cultured based on mature commercial biological products (STEMdiff™ Hematopoietic Kit Catalog # 05310)²⁻⁴. In fact, we realized this mistake after submission, and we appreciate the reviewer's keen insight and guidance. According to the reviewer's suggestions, we have made the corresponding changes in the revised **Supplemental Figure 1C**.

1. My main concern is that what the cell-specific role of *nlr3* is. The expression of *nlr3* seems broad in many cell types, including ECs and HSPCs in zebrafish CHT, and HEC, EC and HSPC in mouse and human AGM. Given that the main finding of the manuscript, i.e., *nlr3* is a regulator for HSPC emergence in zebrafish, I wondered what is the expression pattern of *nlr3* in the AGM region in zebrafish, but not in the zebrafish CHT as shown in Figure 1A by the authors, and what's the EC, HEC or HSPC specific regulatory role of it?

Answer: To address the question of *nlr3* expression, we employed a specific probe synthesized from the full-length mRNA of *nlr3* and conducted WISH experiments on zebrafish wild-type embryos at different developmental stages (**Supplemental Figure 2A**). The data revealed that *nlr3* expression appeared as early as the 1-cell stage and showed specific expression in the AGM region of zebrafish at the 24-28hpf, which coincides with the onset of HSPC generation. Interestingly, at 72hpf in the CHT region, there was also a high expression of *nlr3*, which follows a similar spatiotemporal expression pattern as HSPC generation, indicating that *nlr3* may be involved in the development of HSPCs.

To better distinguish the expression of *nlr3* in EC or HEC, we used *dlc* and *efnb2* probes to label arterial vessels and performed co-expression experiments with the *nlr3* probe by WISH. The results showed a significant enhancement of staining in the arterial part, indicating that *nlr3* is prominently expressed in the endothelial cells of the arterial region in the AGM (**Supplemental Figure 2B, 2C**).

Subsequently, we performed WISH experiments using specific probes for HSPCs, *runx1* and *cmyb*, on zebrafish embryos at 28hpf and 36hpf. Co-expression with *nlr3* probe enhanced the expression of HSPCs, indicating that *nlr3* has temporal and spatial consistency with HSPCs in zebrafish (**Supplemental Figure 2D, 2E**).

Moreover, it should be noted that WISH cannot accurately distinguish between EC, HEC or HSPC. Considering the widespread role of pattern recognition receptors, the broad expression of *nlr3* in various cell types is understandable. Our study, however, primarily focuses on the functional role and mechanisms of *nlr3* in

regulating the generation and development of HSPCs. We believe that with the gradual application of effective techniques such as spatial transcriptomics based on zebrafish, particularly those related to the hematopoietic system (which has not yet been reported), more evidence at the tissue structure level will be presented in the future.

Supplemental Figure 2. The expression of *nlrc3* pattern in zebrafish.

A. Expression pattern of *nlrc3* during zebrafish embryogenesis: The stage examined by whole-mount in situ hybridization (WISH) is shown in each panel: 1-cell stage, 4-cell stage, 12 hours postfertilization (hpf), 20 hpf (Scale bars: 200 μ m), 25 hpf, 36 hpf and 72 hpf. B. Expression of arterial marker *efnb2* in control embryos and control embryos with co-expression of *nlrc3* probe at 28 hpf by WISH. C. Expression of arterial marker *dlc* in control embryos and control embryos with co-expression of *nlrc3* probe at 30 hpf by WISH. D. Expression of HSPC marker *runx1* in control embryos and control embryos with co-expression of *nlrc3* probe at 26 hpf by WISH. E. Expression of HSPC marker *cmyb* in control embryos and control embryos with co-expression of *nlrc3* probe at 36 hpf by WISH. Scale bars, 100 μ m in B-E.

2. Since HSC emergence is fine-tuned and controlled by inflammatory signaling, not only NF- κ B and notch signaling, but also by the primitive myeloid cell and their secreted cytokines. For example, primitive neutrophils secrete *Tnfa* to activate NF- κ B and notch signaling to initiate HSC generation (PMID: 25416946); primitive macrophages also dynamically interact with emerging hematopoietic cells and enhance HSC generation etc (PMID: 25686881). In this manuscript, the authors characterized that the primitive neutrophils and macrophages were also reduced in *nlr3*-deficient embryos, would it be possible that the observed HSPC generation defect was a secondary effect caused by the reduced primitive cells? What's the relationship of primitive immune cells with these HEC/HSC defect?

Answer: The two papers mentioned by the reviewer are of great significance, and the reviewer's suggestions are also very ingenious. The regulatory relationship between the primitive and definitive waves is still to be studied. Although these waves are distinguished by time and space and are considered to be two distinct waves of blood cells, we believe that they are closely related. Moreover, the various types of cells in primitive wave also possess unique functions, such as primitive macrophages. As reported by Kissa, K. et al., the physical conditions for the growth of HSCs are created by primitive macrophages.⁵ “*these tiny entry points are created by macrophages during their migration through the stroma, which will allow HSPCs to join the PCV.*”

On the other hand, primitive macrophages contribute to the differentiation of various cell types,⁶ which goes beyond the realm of simple immune function or even the scope of the hematopoietic system. Currently, this regulatory network appears to be highly complex, and further omics-based investigations and functional validations will reveal additional insights.

Indeed, we have also observed that *il6* derived from primitive neutrophils regulates the generation of HSPCs.⁷ In this study, we believe that in zebrafish, *nlr3* is distinct from inflammatory factors such as *tnfa* and *ill*⁸, and is a constituent of innate immunity. As mentioned earlier, its expression even predates the appearance of the primitive wave. It is unlikely that *nlr3* is secreted by the primitive wave. Furthermore, upon analysis of the RNA-seq data, as shown in the **Figure 4C**, we found that inflammatory factors such as *tnfa*, *ill*, *ifng* did not show statistically significant decreases except for *il6*. As the sequencing was performed during the EHT stage, the source of these factors can still be traced back to the primitive wave. Therefore, this result to some extent indicates that the regulatory role of *nlr3* in the primitive and definitive waves may not be closely related. The viewpoint raised by the reviewer is intriguing and worthy of further exploration, although we believe it does not align with the content of our manuscript.

3. The similar regulatory mechanism of *nlr3* in regulating HSC generation and differentiation was not convincing as they described in Supplemental Figure 8C and was not fully supported with sufficient data, see page 16, 1st paragraph.

Answer: Thanks for the reviewer's suggestion. In order to investigate the restorative effect of *hey1* on *nrc3* deficiency in various blood cell lineages, we employed Sudan Black to observe myeloid cells⁹ and *gata1a* probes to observe erythrocytes. The results of WISH and qPCR revealed that, in comparison to the control group, the overexpression of *hey1* significantly enhanced the downregulation of myeloid cells and erythrocytes caused by *nrc3* deficiency or early Notch inhibition (**Supplemental Figure 9A-C**).

A. Expression of *Sudan Black* in control embryos, treatment group with DAPT, treatment group with DAPT and with overexpression of *nrc3* mRNA, treatment group with DAPT and with overexpression of *hey1* mRNA at 96 hpf. **B.** Expression of *gata1a* in control embryos, treatment group with DAPT, treatment group with DAPT and with overexpression of *nrc3* mRNA, treatment group with DAPT and with overexpression of *hey1* mRNA at 96 hpf.

4. Moreover, since *nrc3* was expressed in both EC and HSPC in the CHT as the Figure 1A shown above, how to rule out the cell extrinsic regulation on HSPC differentiation in the CHT, for example, from the CHT ECs pockets?

Answer: The idea proposed by the reviewer is intriguing. The article describing the concept of CHT ECs pockets sheds new light on the role played by perivascular niche

represented by endothelial cells, and has opened up new avenues for research about colonization and expansion of HSPCs in FL.¹⁰ However, our manuscript mainly focused on the EHT stage, and the data obtained during the CHT stage were aimed at illustrating the changes that occur in HSPCs derived from EHT when they migrate to the CHT stage, in order to demonstrate that EHT is indeed affected.

In fact, we also imaged Tg (*kdrl:mCherry*) cells in the CHT region at 72hpf and found no significant changes due to the absence of *nrc3* (**Figure R1**). We also conducted qPCR analysis, which revealed no statistically significant differences in endothelial-related markers in the CHT tissue samples (**Figure R2**). All in all, the expression and function of *nrc3* in blood vessels may differ from its role in HSPCs generation.

Figure R1. Confocal imaging showing the CHT of Tg (*kdrl:mCherry*) in control embryos and *nrc3* morphants at 72 hpf.

Figure R2. Expression of endothelial and arterial gene *hey2*, *ephrinB2*, *kdrl*, and *deltaC* in CHT of control embryos and *nrc3* mutants at 72 hpf by qPCR.

5. Nlrc3 conditional KO or overexpression approaches specifically in ECs or hematopoietic cells, should be considered for determining the exactly regulatory role of Nlrc3 during HSPC development.

Answer: We appreciate the reviewer's suggestion. However, due to technical and time constraints, we were not able to use more advanced conditional knockout animal models, and we cannot rule out the potential impact of other factors caused by the global knockout. On the other hand, as shown in **Figure 7A-C**, by sorting CD31-marked endothelial cells from the AGM region and examining the phenotype of HSPC-related markers after in vitro culture, we have provided evidence to some extent that the regulatory role of *Nlrc3* has hematopoietic specificity. Moreover, we have already established a model by crossing the global knockout mice with Vav-cre mice which is specific to the hematopoietic system, and future research will continue to build on this model for further investigation.

6. Since the Notch signaling is dynamic from artery/vein specification to HEC/HSPC determination, its activity timing should be considered in the rescue experiments.

Answer: We appreciate the reviewer's suggestion. As suggested, we have taken note of the dynamic regulation of Notch signaling since it has been reported that the activation of Notch signaling before arteriovenous differentiation can promote the production of hematopoietic stem cells. After arteriovenous development is completed, the sustained activation of Notch signaling in arterial endothelial cells can suppress the production of hematopoietic stem cells. Lowering the strength of the Notch signal, on the other hand, can promote the production of hematopoietic stem cells.¹¹

Therefore, in our heat shock experiments, the activation time point was set at 18 hpf. The Notch-specific inhibitor DAPT was treated with embryos at 12 hpf at a concentration of 100 μ M as previously reported^{7, 12} (**Figure 5A-B**).

Minor concerns:

1. The title of Figure 1 was not consistent with the data in the figure, and was exactly the same to sFigure1.

Answer: We thank this reviewer for pointing out this issue. We have corrected the title in the revised manuscript.

Reviewer #2 (Remarks to the Author):

The manuscript titled "Nlrc3 signaling is indispensable for hematopoietic stem cell emergence via Notch signal in vertebrates" describes a comparative approach for addressing factors governing HSPC emergence and

*differentiation in zebrafish and mice. The authors leveraged the utility of the zebrafish model to address the role(s) of a specific pathogen recognition receptor gene for potentially activating Notch governed pathways leading to emergence of myeloid and erythroid lineage development. By using combinations of *nlrc3* mutants and morphants, whole mount in situ hybridization, gene expression analyses coupled with external scRNAseq data, the authors demonstrated a unique role for NLRC3 for HSC and PC development in zebrafish. They then used this information to see if *nlrc3* had an evolutionarily conserved role for HSPCs. Using appropriate murine models coupled with RNAseq, flow cytometry and bone marrow transplantation, the authors provide compelling evidence that the role of *nlrc3* for HSPCs is conserved for vertebrates and thus presents a novel finding. This reviewer had some difficulty following the developing story about *Nlrc3* owing to the data-dense results section that lacked proper definition of genes and associated markers used for delineating the role of *Nlrc3*. This is important as the manuscript should be accessible to a wide readership. Below the authors will find comments and suggestions that are meant to improve the overall accessibility of the results for broad readership.*

Answer: We thank the reviewer for his/her positive general remarks on our manuscript. We appreciate the valuable feedback from the reviewer, and we agree that we did not provide adequate description for the numerous strains. According to the reviewer's suggestions, we have made the corresponding changes in the revised manuscript.

*Major comment: The study utilized a variety of different transgenic zebrafish for demarking different primitive and definitive cell types but rarely were there proper definitions/descriptions for the specific transgenics so that a reader from a related field of study could follow the emerging story without having to look them up. For example, use of the *kdr1* transgenic fish was first mentioned on page 6 but it was not until page 11 that the function of *kdr1* was described—e.g. that *kdr1* is the receptor for VEGF. Reader should not have to go to supplemental section to know what these markers represent for HSPC development. A table describing all transgenics used in this study would be useful.*

Answer: We thank this reviewer for pointing out this important issue. Therefore, we not only added a description of the transgenic strains and the cell types they represent in the manuscript, but also compiled a table in the **Supplementary Table S3**.

General comment about the results section: It was not always clear if the authors were discussing results from morphants or mutant zebrafish. The authors used morpholinos for the Tg fish, and they should clearly state this. Very little discussion follows the results section and therefore the manuscript

could benefit from a combined results/discussion style with a concluding statement at the end.

Answer: Thanks for reviewer's valuable feedback. In general, we used morphants for fluorescence fish experiments, while WISH and qPCR mainly involved mutants. In the figure, the label "MO" refers to morphant, while the label "-/-" refers to mutant. In the rescue experiment, we used mutant strains to consider embryonic tolerance. In response to the reviewer's request, we have provided more detailed descriptions and explanations of these details in the revised manuscript.

Minor:

For the purposes of review, it would have been very helpful to have numbered lines and pages.

Answer: We appreciate the reviewer's suggestion. As suggested, we have added the information in the revised manuscript.

The zebrafish genome is known to encode multiple Nlrc genes—can the authors mention how many if zebrafish code for Nlrc3 paralogs and that they Nlrc3 gene studied in this manuscript is orthologous to human Nlrc3.

Answer: Thanks for the reviewer's comment. Indeed, the NLR is a large family containing numerous paralogues. In our single-cell RNA sequencing analysis of **Figure 1**, we compared the expression of these paralogues in the blood system. However, during the bioinformatics analysis, many paralogs with low expression levels will be excluded, and what we present are the paralogs that have statistical significance. We have added this information in the revised manuscript of Line7 Page5.

Page 2, line 20: citations 10-12 are not for teleosts

Page 3, lines 1-3: There is a reference for Hey2 but not for Hey1 nor is there any description what Hey gene products do—GeneCards indicates that these are helix-loop-helix transcription factor—that would be good to know.

Page 3: line 8. Nlrc3 instead of Nlcr3.

Page 3, line 14: authors should cite the “including published data”

Page 4: EHT was defined in the abstract, but it should be redefined in the main text.

Answer: We thank this reviewer very much for listing out all these editing issues. We have revised all the above-mentioned points in the revised manuscript to ensure the accuracy and clarity.

Page 3: Line 16: “Mechanistically...Hey1”. Qiao et al 2012 and Liu et al. 2017 also suggested that Nlrc3 might be regulating actions of NfKB.

Answer: We appreciate the reviewer's suggestion. And we have added these papers referred to by the reviewer (PMID: 22569257, PMID: 28215032).

Page 4: line 10: "And the expression...HSPC development". Nlrc3 and Tlr4 expression have similar expression levels in mice, but it does not seem to hold true for zf and humans. Figure 1 indicates that zf Nlrc6 is expressed at relatively high levels but not mentioned—is there any information on potential roles of Nlrc6 for HSPCs? On a similar topic, NLRs interact with ASC and Caspase to form inflammasomes when acting as cytosolic PRRs but nothing is said about ASC or CaspA/B expression during HSPC development or how the inflammasome itself could influence lineage development. Do the authors suspect that a similar phenotype would be observed for zebrafish Asc^{-/-} fish (Li et al. 2020) as that for the Nlrc3 mutant and morphants?

Answer: Thanks for the reviewer's reminder. With regards to the expression of *Nlrc3* and *Tlr4*, reports on the regulation of hematopoietic stem cell development and generation by various inflammatory factors and pattern recognition receptors are very limited. We have included most of the relevant studies in our data mining, which have been conducted in zebrafish and mice, and we have tried to select bioinformatics databases that include these species. But the differences between species cannot be ignored. Moreover, instead of solely focusing on the expression level, we are more interested in the downstream molecular mechanisms by which these genes exert their effects, as well as the regulatory networks that they are involved in.

In fact, we have generated a variety of zebrafish mutants with knockouts of the entire NLR family, including *NLRC6*, *ASC*, and *CASP*, in order to conduct a large-scale screening. However, in consideration of the independence of this study, we did not include the content of other family members in this manuscript, although we are conducting related studies.

Question: Could there be a role for STING in regulating NLRC3 during different stages through HSPC development? Just curious owing to the involvement of STING during NLRC3/NFKB mediated responses.

Answer: Thanks for the significant input from the reviewer. As demonstrated in the updated heatmap **Figure 4C**, we found the downregulation of *sting* and *cGAS* expression in our sequencing data, and *cGAS* has statistical significance in our results, suggesting that it may be downstream of *nlrc3*. However, as mentioned in our previous response, we did not delve into this topic in this manuscript. Considering the critical role of *cGAS/STING* in the field of innate immunity, our research team has generated relevant mutants, and research work is still ongoing.

Figure 1A: There are two genes for zf TNF and IFNG—paralogs should be included especially since their expression is associated with Notch regulating HSPCs.

Answer: Thanks for the reviewer's suggestions. We have incorporated TNF paralogs, including *tnfa* and *tnfb*, into F1A, while IFNG was not detected as shown in the R language interface. Therefore, we have included *ifngr1*, *ifngr1l* and *ifngr2*.

```
grep(rownames(sample_FL@assays$RNA@counts), pattern = "^ifn", value = T)
```

```
'ifngr2' · 'ifnphi3' · 'ifnphi1' · 'ifngr1' · 'ifngr1l'
```

The bubble plot is shown below or can be seen in the revised manuscript of **Figure 1A**.

Page 5: “was highly correlated”, Sup figure 1B. This implies regression analysis, but I did not see that in the supplemental figure or in methods. Was regression analysis a part of the bubble plot and hierarchical clustering?

Answer: Thanks for the reviewer's correction. Our description was not accurate. In fact, we performed pairwise comparison analysis based on differential gene expression obtained from RNA-seq, which belongs to T-tests. We have already corrected the description in the revised manuscript to "potential associated".

The authors conducted a very thorough study that implicates *Nlrc3* as a contributor for HSPC development and proposed a signaling/regulatory cascade—the manuscript might benefit from a graphical description of the proposed “NFκB-NLRC3-NOTCH-HEY1” signaling pathway for the reader so that they can better appreciate the pivotal role of NLRC3.

Answer: We appreciate the reviewer’s suggestion. In fact, we have already drawn a hypothetical diagram in **Supplement Figure 10H**.

NICD needs to be described—readers from other disciplines would have no idea what this represents.

Figure 4B, three dimensional PCA is hard to appreciate—PC3 (0.02%, very low) could be eliminated from the plot

Figure 4 legend: The authors need to adjust their A-F lettering. There is no F.

Answer: We thank a lot for the reviewer’s detailed corrections, and we have incorporated all the changes and additions in the updated manuscript.

Reviewer #3 (Remarks to the Author):

The manuscript investigates the Oligomerisation (NOD) like receptor 3 in endothelial-to-hematopoietic transition (EHT) and HSPC emergence. EHT and HSPC emergence are known to be pivotally regulated by the Notch pathway, which in this context is induced amongst other by inflammatory signals and NF-

kB. NLRC3 was previously shown to express in various immune cells and its modulation to affect proinflammatory cytokines.

*Here, the authors suggest *nlrc3* as an upstream activator of Notch activity and the Notch downstream *hey1* gene, and NF- κ B signaling as an upstream inducer of *nlrc3* transcriptional activity.*

*Various expression analyses in three species (zebrafish, mouse and human) relying amongst other on published data and on own data from mouse/human differentiating embryonic stem cells, indicate that *nlrc3* is expressed during development in endothelial and early hematopoietic cells. Elaborate genetic and rescue experiments performed in zebrafish convincingly position *nlrc3* as an upstream regulator of Notch activity required for HSPC emergence. *Nlrc3* appears to mediate its effects on the Notch pathway especially by upregulating *notch1a* and *hey1* expression, which when overexpressed could partially compensate the loss of *nlrc3* during definitive HSPCs production. In the primitive wave of hematopoiesis, loss of *Nlrc3* affected primitive myeloid but not primitive erythroid cells, indicating that it perhaps plays a separate (yet unexplored) role in the myeloid lineage.*

*Importantly, inhibition of NF- κ B affected *Nlrc3* expression, but *nlrc3* expression could not rescue inhibition of NF- κ B, thereby positioning NF- κ B upstream of *nlrc3*.*

*While the manuscript convincingly places *nlrc3* “between” NF- κ B and Notch by various experiments performed in three species and different experimental models, it does not address in any way and in any model the interaction with other major regulators of Notch and hematopoiesis at this precise stage of development such as *vegf*, *evi1*, *shh*, *wnt16*, *gata2b* (see also following manuscripts: PMID: 27638855, PMID: 12110173, PMID: 21654806, PMID: 15737934, PMID: 27015586 and others).*

Answer: We appreciate the reviewer’s thoughtful suggestion. We had designed rescue experiment and provided response and description of it in the following text. And appropriate references have been included in the revised manuscript.

Major comments:

*1.) See also major comment above. Is the expression of *vegf*, *evi1*, *shh* affected by loss of *nlrc3*, and respectively does loss of *nlrc3* affect the expression of *vegf*, *evi1* or *shh*? Are *vegf*, *evi1*, *shh* required for *nlrc3* to mediate its effects on HSPC development? Can notch be induced by *vegf* or *evi1* in the absence of *nlrc3* and can such induction compensate for the loss of *nlrc3*? Do these pathways interact, are they in parallel required to induce HSPCs from endothelial cells?*

*Given the multitude of data produced around the NF- κ B => *Nlrc3* => Notch axis, it is hard to understand why these other major regulators of exactly this step are completely experimentally ignored, and not even adequately discussed.*

Answer: We thanks to the reviewer whose suggestion regarding the regulators of Notch is of great significance to the quality of this manuscript. To address this, we designed a specific rescue experiment. Firstly, we analyzed our RNA-Seq data and found that the VEGF family genes, *vegfaa* and *vegfab*, were significantly down-regulated, as well as *wnt16* and *evil*. *Shha* and *shhb* showed no statistically significant difference in expression (**Figure 4C**). Therefore, we synthesized a full-length mRNA overexpression system in zebrafish containing *wnt16*, *evil*, *vegfaa*, and *vegfab*. For *vegfaa*, which has two transcripts, we selected the longer transcript, NM_131408.3 from NCBI Reference Sequence, to prevent loss of information. Similarly, for *vegfab*, which also has two transcripts, we synthesized the longer transcript, NM_001328597.1, to ensure that all relevant information was included.

After synthesizing the full-length mRNA containing *wnt16*, *evil*, *vegfaa*, and *vegfab* in zebrafish, we overexpressed these genes in *nlrc3* mutants and performed qPCR analyses to preliminarily confirm the expression efficiency and the expression of hematopoietic-related markers *runx1* and *cmyb* (**Supplemental Figure 8A-B**). The qPCR results showed that both *evil* and *vegfa* family genes did not effectively rescue the hematopoietic defects caused by *nlrc3* mutation, but *wnt16* achieved a rescuing effect (**Supplemental Figure 8C**). Therefore, we performed WISH at 28hpf and 36hpf during the EHT stage. The results of WISH also confirmed this conclusion. We have added these important results in our manuscript because they provide a new perspective on the relationship between *nlrc3* and Notch. Specifically, *wnt16* may be acting as a regulator between *nlrc3* and Notch.

Supplement Figure 8. Nlrc3 regulated HSPC production via activating the Notch signaling.

A. Expression of HSPC marker *runx1* in control embryos, *nlrc3* mutants, *nlrc3* mutants with overexpression of *wnt16* mRNA at 28 hpf by WISH. B. Expression of HSPC marker *cmyb* in control embryos, *nlrc3* mutants, *nlrc3* mutants with overexpression of *wnt16* mRNA at 36 hpf by WISH. C. Expression of *wnt16*, *runx1* and *cmyb* in control, control with overexpression of *wnt16* mRNA, *nlrc3* mutants, and *nlrc3* mutants with overexpression of *wnt16* mRNA at 28 hpf by qPCR.

2.) How is the expression of *gata2b* affected by loss of *nlrc3* or *nlrc3* overexpression? Can *gata2b* overexpression rescue hematopoiesis in *nlrc3* morphants or mutants?

Answer: Thanks for the reviewer's suggestion. As described above in the text, our RNA-Seq data also indicated downregulation of *gata2b*. Therefore, we synthesized the full-length mRNA of *gata2b* and overexpressed it. WISH showed that *runx1* and *cmyb* were partially rescued, and qPCR confirmed similar results (**Supplemental Figure 3F-H**).

F. Expression of HSPC marker *runx1* in control embryos, *nlrc3* mutants, *nlrc3* mutants with overexpression of *gata2b* mRNA at 28 hpf by WISH. **G.** Expression of HSPC marker *cmyb* in control embryos, *nlrc3* mutants, *nlrc3* mutants with overexpression of *gata2b* mRNA at 36 hpf by WISH. **H.** Expression of *gata2b*, *runx1* and *cmyb* in control, control with overexpression of *gata2b* mRNA, *nlrc3* mutants, and *nlrc3* mutants with overexpression of *gata2b* mRNA at 28 hpf by qPCR.

3.) *Notch regulates EHT, but at which stage and in which cell type does nlrc3 precisely induce its effects? This is not fully elucidated, and some results are presented in a confusing manner.*

Figure 1 shows analysis of published data bases across three species indicating that nlrc3 is expressed in endothelial and early hematopoietic cells. Expression is found in both venous and arterial endothelial cells, it is unclear from these data at what stage nlrc3 may regulate HSC emergence. What is the role of nlrc3 in venous cells, is there potentially a separate role than the here delineated role in HSPC development,

Answer: We express our gratitude to the reviewer for the valuable feedback. In the revised data, we have included the in-situ expression experiment of *nlrc3*. Using WISH, we discovered that *nlrc3* begins to express in the one-cell stage of zebrafish and is highly expressed during the EHT stage at 26-36hpf. *Nlrc3* continued to express in the DA region representing EHT until the CHT region representing the expansion of HSPCs (**Supplemental Figure 2A**). By co-staining with arterial endothelial-specific markers *dlc* and *efnb2*, as well as hematopoietic stem cell-specific markers *runx1* and *cmyb*, we found that *nlrc3* can effectively enhance the expression of HE and HSPCs (**Supplemental Figure 2B-E**). These results suggest that *nlrc3* may be a sufficient condition for EHT in zebrafish.

Considering *nlrc3* as part of the innate immune system, we do not find its extensive expression within veins to be unacceptable. Interestingly, upon analysis of sequencing data, we have detected the expression of the venous marker *flt4*, which did not appear to be down-regulated due to the absence of *nlrc3* (**Figure 4C**). This finding indicates that the differences in expression levels observed between the physiological state and the knockdown condition may be attributed to varying dosage requirements. While the idea of venous expression being involved in the regulation of HSPC generation is intriguing, it is not the primary focus of this manuscript. We will endeavor to investigate this concept in future studies.

Supplemental Figure 2. The expression of *nlrc3* pattern in zebrafish.

A. Expression pattern of *nlrc3* during zebrafish embryogenesis: The stage examined by whole-mount in situ hybridization (WISH) is shown in each panel: 1-cell stage, 4-cell stage, 12 hours postfertilization (hpf), 20 hpf (Scale bars: 200 μ m), 25 hpf, 36 hpf and 72 hpf. B. Expression of arterial marker *efnb2* in control embryos and control embryos with co-expression of *nlrc3* probe at 28 hpf by WISH. C. Expression of arterial marker *dlc* in control embryos and control embryos with co-expression of *nlrc3* probe at 30 hpf by WISH. D. Expression of HSPC marker *runx1* in control embryos and control embryos with co-expression of *nlrc3* probe at 26 hpf by WISH. E. Expression of HSPC marker *cmyb* in control embryos and control embryos with co-expression of *nlrc3* probe at 36 hpf by WISH. Scale bars, 100 μ m in B-E.

Fig. 1A: shows nlr3 expression in cells from the caudal hematopoietic tissue. However, HSCs are formed from specialized endothelial cells (ECs) in the ventral wall of the dorsal aorta (VDA), within the AGM region. These early blood stem cells subsequently migrate to the caudal hematopoietic tissue, which

supports HSC expansion, before HSCs finally home to the kidney marrow. Thus, nlrc3 expression in cells of the CHT would mark expression in cells in which hematopoietic fate has already been established. This is not as helpful for the here proposed role of nlrc3 in HSPC emergence, please comment.

Answer: We thank for reviewer's insightful comment. As suggested, we analyzed the existing single-cell sequencing data that has been reported for the AGM region in mammals. However, we were unable to find any single-cell sequencing data for the AGM region in zebrafish. In fact, we have been trying to fill this gap for several years by using techniques such as **Figure 2A** dual-fluorescence cell sorting to obtain enough viable cells for library preparation and sequencing. Unfortunately, we have faced numerous technical challenges, and have not yet achieved satisfactory results. Moreover, with the increasing availability of single-cell sequencing data for mammalian cells, the importance of fish data seems to be diminishing.

Therefore, we were only able to perform RNA-seq on *fli1a*:EGFP-labeled cells at the EHT stage, which were sorted from 26hpf embryos. These cells represent the vascular cells of the AGM region, and we hope that this data can provide insight for future validation experiments. Similar strategies have been reported recently.¹³ We also explored single-cell sequencing for these cells and have successfully prepared libraries, which will be presented in our next work.

Fig. 1B. Mouse cells are compared (Aorta Gonad Mesonephros (AGM)-derived HSPCs, fetal liver (FL)-derived HSPCs, venous endothelial cells (VECs), arterial endothelial cells (AECs), pre-hemogenic endothelial cells (pre-HECs) and hemogenic endothelial cells (HECs). Here the authors conclude that "the expression of Nlrc3 increases with hematopoietic maturation, especially when comparing HECs with HSPCs, which is the stage that EHT occurs." When looking at the results, it remains however again unclear at which stage exactly nlrc3 should exert its activity. Puzzling is for example its expression in arterial as well as venous endothelial cells (AECs, VECs) in the same plot. Fig. 1C: shows expression in all analyzed human cell types. Supplemental Figure 1 further compares expression of Nlrc3 in fetal liver HSPCs, bone marrow HSPCs and in cells from differentiating mouse and human embryonic stem cells. A parallel is drawn between Nlrc3 and Tlr4 and IFN- γ expression, since these genes are also involved in HSC emergence.

Answer: We appreciate the reviewer's reminder, and we agree that although single-cell transcriptome sequencing is of great significance, it still has limitations in accuracy and timeliness. Therefore, we believe that spatial transcriptomics may be an effective means of studying specific expression patterns. Unfortunately, we were unable to obtain useful information from the analysis of this zebrafish data, as the sequencing time point ended at 24hpf in this article,¹⁴ and the EHT had not yet fully started. Furthermore, to date, reports on spatial transcriptomics related to zebrafish blood have not been seen. In fact, including the previously mentioned single cells in the EHT stage, this is the direction

we are currently striving to conquer. Perhaps this can reveal why *nlrc3* is expressed differently in various cells, including veins, and its involvement in regulating the HSPC process.

On the other hand, in our manuscript, *nlrc3* is involved in regulating HSPC formation, especially during the EHT stage, which is why its expression increases from AE to HE, and decreases after the end of EHT when HSPCs are formed. The data from the most critical human embryo study confirms this hypothesis. Our functional experiments also provide some evidence to support this claim. In fact, compared with other published pattern recognition receptors, including *tlr4* and RIG-I-like Receptors *rig-I* or *mda5*, *nlrc3* exhibits a similar trend in expression and is even expressed more strongly. Therefore, in general, this study is still based on the specific function of *nlrc3*, and the exploration of single-cell transcriptome data is only to illustrate its potential significance in expression. We believe that functional experiments are more convincing and have practical value.

Overall, these data primarily show that Nlrc3 is expressed in endothelial and hematopoietic cells and that its expression associates with that of other inflammation associated genes. It is unclear in fact, how ubiquitous the expressed of these genes is? It would be more informative to show association/clustering/co-expression with lineage-specific markers, such as markers of endothelial and especially hematopoietic identity previously implicated in EHT (gata2, runx1, cmyb, evi1). These analyses need to be shown.

Answer: We appreciate the reviewer's suggestions. We have conducted rescue experiments and co-expression assays and updated the manuscript accordingly.

Minor comments:

- 1.) *Suppl. Fig. 6: what are the top differentially regulated genes?*
- 2.) *Suppl. 7 title: please correct typo*

Answer: We appreciate the reviewer's suggestions and have updated both the manuscript and figures accordingly.

References

1. Shan, W. *et al.* Enhanced HSC-like cell generation from mouse pluripotent stem cells in a 3D induction system cocultured with stromal cells. *Stem Cell Res Ther* **12**, 353 (2021).
2. McQuade, A. *et al.* Development and validation of a simplified method to generate human microglia from pluripotent stem cells. *Mol Neurodegener* **13**, 67 (2018).
3. Crawford, L.B. *et al.* CD34(+) Hematopoietic Progenitor Cell Subsets Exhibit Differential Ability To Maintain Human Cytomegalovirus Latency and Persistence. *J Virol* **95** (2021).
4. Themeli, M. *et al.* iPSC-Based Modeling of RAG2 Severe Combined Immunodeficiency Reveals Multiple T Cell Developmental Arrests. *Stem Cell Reports* **14**, 300-311 (2020).
5. Kissa, K. *et al.* Live imaging of emerging hematopoietic stem cells and early thymus colonization. *Blood* **111**, 1147-1156 (2008).
6. Ferrero, G. *et al.* Embryonic Microglia Derive from Primitive Macrophages and Are Replaced by *cmyb*-Dependent Definitive Microglia in Zebrafish. *Cell Rep* **24**, 130-141 (2018).
7. Tie, R. *et al.* Interleukin-6 signaling regulates hematopoietic stem cell emergence. *Exp Mol Med* **51**, 1-12 (2019).
8. Frame, J.M. *et al.* Metabolic Regulation of Inflammasome Activity Controls Embryonic Hematopoietic Stem and Progenitor Cell Production. *Dev Cell* **55**, 133-149 e136 (2020).
9. Liu, W. *et al.* *c-myb* hyperactivity leads to myeloid and lymphoid malignancies in zebrafish. *Leukemia* **31**, 222-233 (2017).
10. Tamplin, O.J. *et al.* Hematopoietic stem cell arrival triggers dynamic remodeling of the perivascular niche. *Cell* **160**, 241-252 (2015).
11. Zhang, P. *et al.* G protein-coupled receptor 183 facilitates endothelial-to-hematopoietic transition via Notch1 inhibition. *Cell Res* **25**, 1093-1107 (2015).
12. Sawamiphak, S., Kontarakis, Z. & Stainier, D.Y. Interferon gamma signaling positively regulates hematopoietic stem cell emergence. *Dev Cell* **31**, 640-653 (2014).
13. Lefkopoulos, S. *et al.* Repetitive Elements Trigger RIG-I-like Receptor Signaling that Regulates the Emergence of Hematopoietic Stem and Progenitor Cells. *Immunity* **53**, 934-951 e939 (2020).
14. Liu, C. *et al.* Spatiotemporal mapping of gene expression landscapes and developmental trajectories during zebrafish embryogenesis. *Dev Cell* **57**, 1284-1298 e1285 (2022).

REVIEWER COMMENTS

Reviewer #2 (Remarks to the Author):

The authors have addressed the comments and concerns that were submitted for the first review. This reviewer does not have any further questions or comments.

Reviewer #3 (Remarks to the Author):

The authors have added a significant amount of data, and have answered some of the concerns of the reviewers. I still lack a comprehensive view of how their new data integrates into the current models, and other signals reported to be involved at this developmental step (vegf, evi1, wnt16). The authors do reply in their response letter that they have performed some experiments that were however not conclusive to them, so they chose not to incorporate them. They should please show these data, and the reason they should do so is, because these are very established pathways and even negative data will be intriguing and foster further research.

The authors need to provide a model figure, in which they integrate their findings into the general picture of EHT molecular regulation - what regulates when, what is downstream what is upstream, what is known and where do the new *Nlrc3* findings integrate. Here they need to also mention the players that they do not otherwise mention at this point in their manuscript, and discuss why data are negative or different. There always are possibilities in all cases (such as pathways regulating in parallel a common downstream pathway etc.)- but other data cannot just be ignored if it does not fit into the immediate model.

In general, cells resulting from in vitro differentiating ESCs are rather myeloid progenitors (primitive ones?) than HSCs.

So *Nlrc3* probably plays a role in EHT/HSC development but also in myeloid progenitors. This should be separated in the data discussion, and presented / discussed separately. Does *Nlrc3* also play in more differentiated myeloid cells, which also react to and are important players in inflammation? The data should be evaluated for confounding findings that rather support latter notion, and this aspect should be discussed.

Comments to the text:

Language needs improvement, e.g.:

Page 3 Lines 1-2: "However, as the upstream regulator of these factors, an interaction between *Nlrc3* and Notch remains undocumented."

"NF- κ B was regulated by *Nlrc3* and affected HSPC generation through modulation of transcriptional activity, and that Notch was associated with *Nlrc3* through the downstream gene *Hey1*. "

Page 4, Lines 25-29: "Alternatively, since the hematopoietic differentiation system using embryonic stem cells (ESCs) is an ideal research model that simulates the in vivo embryonic hematopoiesis, in which cells gradually differentiates into the mesoderm, HE, and HSPCs stages, and are accompanied by specific different molecular markers at different stages.^{30, 31} " Check this sentence for the meaning and be aware that in vitro differentiating ESC often mimick rather primitive hematopoiesis and the formation of hematopoietic progenitors, while true HSCs are challenging to obtain in many of these systems.

Page 5, Line 3: “Differential gene expression analysis using RNA-seq revealed that *Nlrc3* expression was potential associated with the expression of genes that have been reported to regulate the emergence of hematopoietic stem cells, including *Tlr4* and *IFN- γ* potential associated (..) – potentially associated ... But: *Tlr4* and *IFN- γ* definitely are genes that do much more than regulate EHT, so they will be expressed also in many other tissues and cells. Co-expression with more hematopoiesis specific genes is rather useful.

Page 5: “The expression of *NLRC3* in HE was significantly increased by approximately 300-fold, and was even 16 higher in *CD43+* and *CD45+* HSPCs (Supplemental Figure 1E).”

- How is *NLRC3* expressed in adult, bone marrow derived myeloid progenitor cells? It could be also expressed and functional in such cells, and the expression analyses results reported here might reflect this.

Page 6 (Lines 3-6): “By co-staining with arterial endothelial specific markers *dlc* and *efnb2* (Supplemental Figure 2B-C), hematopoietic stem/progenitor cell-specific markers *runx1* and *cmyb* (Supplemental Figure 2D-E), we found that *nlrc3* can effectively enhance the expression of HE and HSPCs. These results suggest that *nlrc3* may be a sufficient condition for EHT in zebrafish.”

- please describe the experiments more accurately here: is this the result of an *nlrc3* overexpression experiment? Even so, the conclusion is questionable: even if *nlrc3* (overexpression?) enhanced *dlc*, *efnb2*, *runx1* and *cmyb*, this does not mean that *nlrc3* is “a sufficient condition”, it rather indicates that it is *nlrc3* may be involved in the regulation of this process and have stimulatory effects. The mutants data that follows is rather important to claim necessity.

Responses to the Reviewers' Comments

We sincerely appreciate the reviewers' time and effort in providing constructive critiques for our initial manuscript. Their valuable comments and suggested experiments have greatly contributed to improving its quality.

Based on their constructive comments, especially regarding the presentation and discussion of negative results, as well as the creation of the model diagram, we have included these in the updated images and made corresponding modifications to the manuscript. The detailed point-by-point responses to reviewers' comments are shown as below.

REVIEWER COMMENTS

Reviewer #2 (Remarks to the Author):

The authors have addressed the comments and concerns that were submitted for the first review. This reviewer does not have any further questions or comments.

Reviewer #3 (Remarks to the Author):

The authors have added a significant amount of data, and have answered some of the concerns of the reviewers. I still lack a comprehensive view of how their new data integrates into the current models, and other signals reported to be involved at this developmental step (vegf, evi1, wnt16). The authors do reply in their response letter that they have performed some experiments that were however not conclusive to them, so they chose not to incorporate them. They should please show these data, and the reason they should do so is, because these are very established pathways and even negative data will be intriguing and foster further research.

Answer: Thank you to the reviewer for your thoroughness and comprehensive perspective. We have added relevant negative data to the updated manuscript. As previously mentioned, we overexpressed full-length mRNA of *vegfaa*, *vegfab* and *evils* in *nlr3* mutants, which were genes significantly downregulated due to *nlr3* knockdown. However, through multiple qPCR tests, we did not observe effective rescue of the downregulation of *runx1* and *cmyb* caused by *nlr3* deletion (**Supplemental Figure 9A-C**). We have also discussed this aspect in the updated manuscript.

Expression of *vegfaa*, *runx1* and *cmyb* in control, control with overexpression of *vegfaa* mRNA, *nlr3* mutants, and *nlr3* mutants with overexpression of *vegfaa* mRNA at 28 hpf by qPCR.

Expression of *vegfab*, *runx1* and *cmyb* in control, control with overexpression of *vegfab* mRNA, *nlr3* mutants, and *nlr3* mutants with overexpression of *vegfab* mRNA at 28 hpf by qPCR.

Expression of *evi1*, *runx1* and *cmyb* in control, control with overexpression of *evi1* mRNA, *nlrc3* mutants, and *nlrc3* mutants with overexpression of *evi1* mRNA at 28 hpf by qPCR.

The authors need to provide a model figure, in which they integrate their findings into the general picture of EHT molecular regulation - what regulates when, what is downstream what is upstream, what is known and where do the new *nlrc3* findings integrate. Here they need to also mention the players that they do not otherwise mention at this point in their manuscript, and discuss why data are negative or different. There always are possibilities in all cases (such as pathways regulating in parallel a common downstream pathway etc.)- but other data cannot just be ignored if it does not fit into the immediate model.

Answer: Thank you to the reviewer's suggestions. In the updated manuscript, we have created a model figure representing the zebrafish developmental stages. Through this model figure, we aim to present our innovative findings regarding *nlrc3* during the EHT stage. These findings include the role of NF-κB upstream of *nlrc3*, as well as *nlrc3*'s regulation of EHT through *notch1a* and *hey1*. Additionally, the loss of *nlrc3* leads to the downregulation of *wnt16*, *gata2b*, *vegfaa*, *vegfab*, and *evi1*. However, the results confirm that only *wnt16* and *gata2b* form an effective rescue. Based on the reported activity windows of these genes,¹ such as *wnt16* mainly acting before 25hpf in zebrafish, and *vegf*, as well as *notch*, believed to start acting around 20hpf, we propose that *nlrc3* plays a crucial role even before the EHT and hematopoiesis initiation stages.

Pattern diagram illustrating *nlrc3* regulation of hematopoietic stem cell generation in zebrafish. During the EHT stage in zebrafish, NF- κ B is upstream of *nlrc3*, while *nlrc3* regulates the generation of hematopoietic stem cells through *notch1a* and the Notch target gene *hey1*. Simultaneously, *nlrc3* also influences two critical genes, *wnt16* and *gata2b*, but does not affect hematopoietic stem cell generation through *vegfa* or *evil*.

In general, cells resulting from in vitro differentiating ESCs are rather myeloid progenitors (primitive ones?) than HSCs.

So *nlrc3* probably plays a role in EHT/HSC development but also in myeloid progenitors. This should be separated in the data discussion, and presented / discussed separately.

Does *nlrc3* also play in more differentiated myeloid cells, which also react to and are important players in inflammation? The data should be evaluated for confounding findings that rather support latter notion, and this aspect should be discussed.

Answer: Thank you for the reviewer's suggestions. To address your question, we analyzed publicly available sequencing data from human bone marrow-derived CD34⁺ cells.² As shown in the following figure, in three distinct types defined by CD123 expression of granulocyte-macrophage progenitors (GMPs) derived from common myeloid progenitors (CMPs), we analyzed pattern recognition receptors including *NLRC3*, inflammatory factors, as well as myeloid-promoting cytokines and myeloid genes. The results indicate that *NLRC3* expression within the subpopulations of

myeloid cells in adult bone marrow is enriched with the inflammatory factors *CSF2RB* and *IFNGR2*. This suggests that *NLRC3* serves as an important player in inflammation.

Bubble plot of sequencing data demonstrating the expression of pattern recognition receptors including *NLRC3*, inflammatory factors, myeloid-promoting cytokines, and myeloid genes. Bone marrow-derived CD34⁺ cells defined by CD123 in three distinct types of granulocyte-macrophage progenitors (GMPs), including Bone marrow-derived GMP1-3 (BM-GMPs), and Bone marrow-derived unknown population (BM-UNK).

In addition to the references already cited in the manuscript, *Nlrc3* has been reported to negatively regulate dendritic cell antigen presentation through the p38 signaling pathway.³ According to the latest publicly available reports, the myeloid-specific deletion of *Nlrc3* has been shown to enhance macrophage glycolysis and mitigate sepsis-induced immunosuppression through the mTOR, *Traf6*, and NF-κB pathways. These findings underscore the unique role of *Nlrc3* in myeloid cells' immune tolerance. It's important to note that the functions of *Nlrc3* during embryonic hematopoiesis and in adult stages may differ. While previous literature has placed *Nlrc3* upstream of NF-

KB, our data suggest that NF- κ B acts upstream of *Nlrc3* in regulating hematopoiesis. We have incorporated these insights into the discussion section of our manuscript.

Comments to the text:

Language needs improvement, e.g.:

Page 3 Lines 1-2: *“However, as the upstream regulator of these factors, an interaction between *Nlrc3* and Notch remains undocumented.”*

*“NF- κ B was regulated by *Nlrc3* and affected HSPC generation through modulation of transcriptional activity, and that Notch was associated with *Nlrc3* through the downstream gene *Hey1*. “*

Page 4, Lines 25-29: *“ Alternatively, since the hematopoietic differentiation system using embryonic stem cells (ESCs) is an ideal research model that simulates the in vivo embryonic hematopoiesis, in which cells gradually differentiates into the mesoderm, HE, and HSPCs stages, and are accompanied by specific different molecular markers at different stages.^{30, 31} “ Check this sentence for the meaning and be aware that in vitro differentiating ESC often mimic rather primitive hematopoiesis and the formation of hematopoietic progenitors, while true HSCs are challenging to obtain in many of these systems.*

Answer: Thank you for the reviewer's feedback. We have revised the manuscript as suggested. In fact, a significant portion of our research efforts are focused on the in vitro culture of hematopoietic stem cells, as far as we are aware, articles on in vitro differentiation have not intentionally emphasized whether it is primitive or definitive.^{4, 5} Additionally, differentiation methods can vary between different laboratories. The investigation of in vivo embryonic hematopoiesis is aimed at better understanding the regulatory network governing hematopoietic stem cell formation, which can support advances in ex vivo hematopoiesis. This remains a challenging endeavor, and currently, we are using cells from different differentiation stages as a research platform to accumulate data provide evidence from different perspectives.

Page 5, Line 3: *“Differential gene expression analysis using RNA-seq revealed that *Nlrc3* expression was potential associated with the expression of genes that have been reported to regulate the emergence of hematopoietic stem cells, including *Tlr41* and IFN- γ 15 potential associated (..) ” – potentially associated … But: *Tlr4* and IFN- γ definitely are genes that do much more than regulate EHT, so they will be*

expressed also in many other tissues and cells. Co-expression with more hematopoiesis specific genes is rather useful.

Answer: We appreciate the suggestions from the reviewer and would like to clarify our approach. To identify potential members within the NLR gene family using in vitro sequencing data, we compared these already reported regulatory members, especially pattern recognition receptors represented by *Tlr4*. Our results revealed a significant enrichment of *Nlrc3* and its functionality was experimentally validated.

In response to the reviewer's query, we have also compared NLR gene family members with hematopoietic-related genes, as shown in the figure below. It was observed that *Nlcr3* is indeed relatively enriched among many genes, including *Runx1*, *Tal1*, *Gata2*, *Cd34*, and *Evi1*. As for *Nod1*, *Nod2*, and other NLR family members, we conducted exploratory knockdown experiments, but to be honest, we did not achieve as significant phenotypes as with *Nlrc3*.

In summary, through the analysis of this in vitro data, our aim was to screen potential regulatory genes, especially in comparison with genes already reported. This is a

comparative screening of upstream factors, providing evidence for functional-level experimental validation.

Page 5: “The expression of *NLRC3* in HE was significantly increased by approximately 300-fold, and was even 16 higher in CD43+ and CD45+ HSPCs (Supplemental Figure 1E).”

- How is *NLRC3* expressed in adult, bone marrow derived myeloid progenitor cells? It could be also expressed and functional in such cells, and the expression analyses results reported here might reflect this.

Answer: As mentioned earlier in the text, we analyzed publicly available sequencing data from human bone marrow-derived CD34⁺ cells.² In three distinct types of granulocyte-macrophage progenitors (GMPs) derived from common myeloid progenitors (CMPs), we analyzed pattern recognition receptors including *NLRC3*, inflammatory factors, as well as myeloid-promoting cytokines and myeloid genes. The results indicate that *NLRC3* expression within the subpopulations of myeloid cells in adult bone marrow is enriched with the inflammatory factors *CSF2RB* and *INFR2*. This suggests that *NLRC3* serves as an important player in inflammation.

Page 6 (Lines 3-6): “By co-staining with arterial endothelial specific markers *dlc* and *efnb2* (Supplemental Figure 2B-C), hematopoietic stem/progenitor cell-specific markers *runx1* and *cmyb* (Supplemental Figure 2D-E), we found that *nlrc3* can effectively enhance the expression of HE and HSPCs. These results suggest that *nlrc3* may be a sufficient condition for EHT in zebrafish.”

- please describe the experiments more accurately here: is this the result of an *nlrc3* overexpression experiment? Even so, the conclusion is questionable: even if *nlrc3* (overexpression?) enhanced *dlc*, *efnb2*, *runx1* and *cmyb*, this does not mean that *nlrc3* is “a sufficient condition”, it rather indicates that it is *nlrc3* may be involved in the regulation of this process and have stimulatory effects. The mutants data that follows is rather important to claim necessity.

Answer: Thank you for the reminder. In order to verify whether the expression of *nlrc3* exhibits spatiotemporal specificity during hematopoiesis, we conducted whole-mount in situ hybridization (WISH) experiments on wild-type embryos. These experiments were divided into two groups: the first group included single staining for vascular markers *dlc* and *efnb2*, as well as hematopoietic markers *runx1* and *cmyb*. The second group involved co-staining with *nlrc3* probe in addition to the aforementioned probes, and the results from these two groups were compared. The findings indicated an increased expression in the co-staining with *nlrc3* groups, suggesting the potential

involvement of *nlr3* in hematopoiesis during the EHT stage. We have incorporated these modifications into the updated manuscript.

References

1. Loeffler D, Kokkaliaris KD, Schroeder T. Wnt to notch relay signaling induces definitive hematopoiesis. *Cell Stem Cell* **9**, 2-4 (2011).
2. Buenrostro JD, *et al.* Integrated Single-Cell Analysis Maps the Continuous Regulatory Landscape of Human Hematopoietic Differentiation. *Cell* **173**, 1535-1548 e1516 (2018).
3. Fu Y, *et al.* NLRC3 expression in dendritic cells attenuates CD4(+) T cell response and autoimmunity. *EMBO J* **38**, e101397 (2019).
4. Xu Y, *et al.* A synthetic three-dimensional niche system facilitates generation of functional hematopoietic cells from human-induced pluripotent stem cells. *J Hematol Oncol* **9**, 102 (2016).
5. Wang D, *et al.* SETD7 promotes lateral plate mesoderm formation by modulating the Wnt/beta-catenin signaling pathway. *iScience* **26**, 106917 (2023).

REVIEWERS' COMMENTS

Reviewer #3 (Remarks to the Author):

The authors have performed further extensive changes. I have no further comments.